
# A stratospheric prognostic ozone for seamless Earth System Models: performance, impacts and future

Beatriz M. Monge-Sanz[1,2], Alessio Bozzo[3,a], Nicholas Byrne[4], Martyn P. Chipperfield[5,6], Michail Diamantakis[7], Johannes Flemming[7], Lesley J. Gray[1,2], Robin J. Hogan[7,4], Luke Jones[7], Linus Magnusson[7], Inna Polichtchouk[7], Theodore G. Shepherd[4], Nils Wedi[7], and Antje Weisheimer[1,2,7]

[1]Department of Physics, University of Oxford, Oxford, United Kingdom
[2]National Centre for Atmospheric Science, University of Oxford, Oxford, United Kingdom
[3]European Organisation for the Exploitation of Meteorological Satellites, Darmstadt, Germany
[4]Department of Meteorology, University of Reading, Reading, United Kingdom
[5]School of Earth and Environment, University of Leeds, United Kingdom
[6]National Centre for Earth Observation, University of Leeds, United Kingdom
[7]European Centre for Medium-Range Weather Forecasts, Reading, United Kingdom
[a]formerly at European Centre for Medium-Range Weather Forecasts, Reading, United Kingdom

**Correspondence:** B. M. Monge-Sanz
(beatriz.monge-sanz@physics.ox.ac.uk)

**Abstract.**

We have implemented a new stratospheric ozone model in the European Centre for Medium-Range Weather Forecasts (ECMWF) system, and tested its performance for different timescales, to assess the impact of stratospheric ozone on meteorological fields. We have used the new ozone model to provide prognostic ozone in medium-range and long-range experiments,

5 showing the feasibility of this ozone scheme for a seamless NWP modelling approach. We find that the stratospheric ozone distribution provided by the new scheme in ECMWF forecast experiments is in very good agreement with observations, even for unusual meteorological conditions such as Arctic stratospheric sudden warmings (SSWs) and Antarctic polar vortex events like the vortex split of year 2002. To assess the impact it has on meteorological variables, we have performed experiments in which the prognostic ozone is interactive with radiation. The new scheme provides a realistic ozone field able to improve

10 the description of the stratosphere in the ECMWF system, as we find clear reductions of biases in the stratospheric forecast temperature. The seasonality of the Southern Hemisphere polar vortex is also significantly improved when using the new ozone model. In medium-range simulations we also find improvements in high latitude tropospheric winds during the SSW event considered in this study. In long-range simulations the use of the new ozone model leads to an increase in the winter North Atlantic Oscillation (NAO) index correlation, and an increase in the signal to noise ratio over the North Atlantic sector. In our study we

15 show that by improving the description of the stratospheric ozone in the ECMWF system, the stratosphere-tropospheric coupling improves. This highlights the potential benefits of this new ozone model to exploit stratospheric sources of predictability and improve weather predictions over Europe on a range of time scales.



# 1 Introduction

The new emerging generation of seamless Earth System Models (ESMs) needs to be developed in ways that allow accurate
performance in timescales from weather to climate, including seasonal and subseasonal scales. This requires slow evolving pro-
cesses that influence the troposphere, like stratospheric processes, to be realistically included. The links between stratospheric
ozone, polar vortex dynamics, and extreme winter weather over Europe are being increasingly recognised (e.g. Kolstad et al.,
2010; Waugh et al., 2017; Kretschmer et al., 2018). Lack of detail in the description of the stratosphere is also linked to an un-
realistic representation of stratosphere-troposphere coupling, which makes most existing models unable to exploit all potential
sources of predictability deriving from the stratosphere (e.g. Scaife et al., 2016).

The stratospheric ozone layer accounts for 90 % of the total existing atmospheric ozone and plays fundamental roles for
atmospheric processes and for life on Earth. Ozone in this atmospheric region provides a vital shield against harmful ultraviolet
(UV) radiation, preventing the most energetic UV-C and UV-B wavelengths (wavelength bands below 300 nm) from reaching
the Earth's surface. UV radiation is absorbed by ozone in the stratosphere via very exothermic reactions, therefore ozone is
the main player in shaping the vertical profile of temperature in the stratosphere and has a fundamental role in the interactions
between radiation and dynamics in this region and the exchange of air masses with the troposphere. Unlike ozone in the
troposphere, where its influence on physics and dynamics is dwarfed by the influence of other meteorological phenomena, a
realistic distribution of ozone in the stratosphere is essential to correctly model the dynamical behaviour in this region.

Interannual dynamical variability of the polar vortex, in both hemispheres, causes large differences in the amounts of ozone
depletion from year to year. In the Northern Hemisphere (NH), the occurrence of sudden stratospheric warming (SSW) events,
with temperatures in the polar stratosphere experiencing very rapid increases, lead to significantly less Arctic ozone loss than
during cold Arctic years without SSW disturbances (e.g. Monge-Sanz et al., 2011; Solomon et al., 2014; Strahan et al., 2016).
Over Antarctica, the formation of the ozone hole every year is caused by the presence of ozone depleting substances (ODS) in
the atmosphere, but its extent and duration depends on the particular dynamics of the polar vortex each year. The amount of
ozone depletion then feeds back to temperature and winds through radiative interactions. Thus, correctly simulating the amount
of polar stratospheric ozone depletion during late winter/spring and allowing it to interact with radiation has also the potential
to improve the way models reproduce stratosphere-troposphere coupling.

Stratospheric ozone research has been very active during the past 30 years (WMO, 2019); however, modelling tools required
to answer remaining and emerging questions at different timescales regarding links between ozone, climate change and weather
extremes, are not yet available. New seamless Earth System Models (ESMs) that integrate climate and weather elements of the
Earth System are starting to be developed and will provide valuable tools to address such questions. How to most efficiently
incorporate appropriate descriptions of stratospheric processes, including stratospheric ozone, is still an open question. Such
descriptions will need to exhibit the right compromise between realism and computational cost to be able to adequately perform
at all timescales.

The Antarctic ozone hole was first discovered in the mid 1980s, and understanding processes regulating the amount and
distribution of ozone in the stratosphere became a high scientific priority for societal needs. For this reason, and to monitor the





effectiveness of the Montreal Protocol, a set of complex atmospheric models was specifically developed to address stratospheric ozone related questions: chemistry-transport models (CTMs) and chemistry-climate models (CCMs) became the best modelling tools to understand links between chemical and dynamical factors governing the formation, distribution and destruction of

stratospheric ozone. Nowadays, these modelling tools include very detailed atmospheric chemistry processes, based on the most up-to-date scientific knowledge, and can provide very accurate simulations of stratospheric ozone (e.g. Eyring et al., 2007; Morgenstern et al., 2017).

Nevertheless, such full-chemistry level of detail is not affordable for high resolution multiple ensemble weather forecasting simulations, because of computational costs. Alternative stratospheric descriptions that are both realistic and affordable for all

time scales are key needs for emerging seamless systems. In this work we assess the feasibility and performance of a linear model for stratospheric ozone that can be implemented in any global circulation model (GCM) within an ESM at very low computational cost, yet providing quality comparable to the ozone field from world leading full-chemistry models.

The first linear ozone model was formulated by Cariolle and Déqué (CD) (Cariolle and Déqué, 1986) when the hetero-

geneous chemistry of the ozone hole was still unknown, and therefore the scheme parameterised only the effects of ozone gas-phase chemistry, ignoring the heterogenous chemistry processes reponsible for polar ozone loss. Susbsequent versions of the CD model kept the initial approach but included an additional term to take into account the polar destruction of ozone at low temperatures (Cariolle and Teyssèdre, 2007). This is the ozone model currently used by the Integrated Forecast System (IFS) of the European Centre for Medium-Range Weather Forecasts (ECMWF). However, this CD approach has significant

limitations in the way it represents heterogeneous ozone loss.

An alternative linear model, hereafter called the BMS model, was more recently developed by Monge-Sanz et al. (2011). The BMS model, as the CD one, is a linear representation of stratospheric ozone sources and sinks as a function of ozone concentrations and temperature. But the BMS model, unlike the CD one or any other previous linear ozone model, consistently includes both gas-phase and heterogenous chemistry for stratospheric ozone. Monge-Sanz et al. (2011) tested this new ozone

model within the SLIMCAT 3D chemistry transport model (CTM) used to obtain the linear scheme, showing the superiority of the BMS scheme over the CD scheme in a multiannual run covering the period 1991-2002. In their study they showed the capacity of the new ozone scheme to provide a stratospheric ozone field of comparable quality to the ozone field from the SLIMCAT full-chemistry model. More details on the differences between the BMS and the CD ozone models are given below in Sect. 2.1, and a full discussion comparing both formulations can be found in Monge-Sanz et al. (2011).

Besides the above mentioned decadal simulations within the SLIMCAT CTM, the BMS scheme has already been adopted by global models for different applications, from numerical weather forecasting to tropospheric air-quality (Jeong et al., 2016; Badia et al., 2017), because of the more realistic simulation of stratospheric ozone it provides, compared to other available options like observation-based monthly climatologies or the CD scheme, for models that cannot afford stratospheric full-chemistry modules.

For the present study we have implemented the new BMS ozone in the ECMWF general circulation model (GCM) and compared the performance of the new BMS and the default CD ozone model schemes in terms of the ozone distributions they



provide. Then we have evaluated the way stratospheric ozone impacts meteorological fields at different timescales. This is the first time that the performance of an ozone model has been assessed for different timescales in a GCM, with the goal of evaluating its feasibility for seamless Earth systems simulations.

The structure of this article is as follows: Section 2 describes the model configuration used and the set of experiments designed for this study, as well as giving an overview of the observations datasets used for validation of our model results. Section 3 shows the ozone distribution results obtained for different experiments, regions and case studies. Then the impacts on meteorological fields are discussed in Sect. 4. The summary of results and conclusions for this study is found in Sect. 5, which also provides a discussion of future work and recommendations.

## 2   Experimental details

### 2.1   New ozone model in the IFS

We have implemented the stratospheric ozone model described by Monge-Sanz et al. (2011), the BMS ozone scheme, within the ECMWF Integrated Forecast System (IFS). The scheme represents the effects of stratospheric ozone sources and sinks following a linear approach, so that a CTM or GCM model can simulate time evolution of ozone by including an advected 100   tracer whose concentration $f$ evolves in time according to the following equation:

$$\frac{\partial f}{\partial t} = c_0 + c_1(f - \bar{f}) + c_2(T - \bar{T}) + c_3(c_{O_3} - c_{\bar{O}_3}), \tag{1}$$

where the coefficients $c_i$ ($i = 0, 1, 2, 3$) are tendencies derived from the full-chemistry CTM runs, and the terms $\bar{f}$, $\bar{T}$, $c_{\bar{O}_3}$ are climatological reference values (in this case obtained from the full-chemistry output fields) for the ozone concentration $f$, 105   temperature $T$ and partial column of ozone above the considered location $c_{O_3}$. The coefficients $c_i$ and the climatological terms are provided as a function of latitude, pressure and month. For complete details on the calculation methodology and CTM runs leading to this ozone model see Monge-Sanz et al. (2011).

    A full discussion on the differences between this new ozone model and the default CD scheme used by the ECMWF operational system can also be found in Monge-Sanz et al. (2011); for completeness we include here below the equation of 110   the default CD scheme, together with a brief description of the most significant differences. The time evolution of the ozone concentration in the default ECMWF scheme is based on the expression

$$\frac{\partial f}{\partial t} = c_0 + c_1(f - \bar{f}) + c_2(T - \bar{T}) + c_3(c_{O_3} - c_{\bar{O}_3}) + c_4(\text{Cl}_{\text{EQ}})^2 f \tag{2}$$

where the tendency coefficients $c_i$ ($i = 0, 1, 2, 3$) include only gas-phase chemistry effects, making it necessary to add the 115   $5^{th}$ term and the corresponding coefficient $c_4$ to account for ozone destruction related to heterogenous chemistry processes. $\text{Cl}_{\text{EQ}}$ is the equivalent chlorine content of the stratosphere, and varies from year to year. This fifth term is only active when temperature falls below 195 K in daytime for latitudes polewards of 45°, and the coefficient $c_4$ is calculated with different





methods and approximations from the rest of the coefficients, reducing the consistency of the approach. Conceptually, this kind of term is too restrictive with respect to the current understanding of heterogeneous processes. For instance, in reality

the temperature threshold for the formation of polar stratospheric clouds (PSCs) actually depends on altitude and trace gas concentrations. Another shortcoming in the heterogeneous approach in Eq. (2) is that it assumes chlorine activation and $O_3$ destruction by sunlight to take place at the same time. Actually, the activation of air masses takes place during the polar night inside the polar vortex, when temperature is low enough. Later, when spring sunlight returns to polar latitudes, the processed air is able to destroy ozone. However, the destruction can also happen during winter if activated air masses reach lower latitudes

(e.g. by filamentation or vortex break-up) and are exposed to sunlight. This latter kind of process is completely missed by a localised temperature term like the one in the default ECMWF scheme.

The treatment of heterogeneous chemistry is one fundamental difference in the new (BMS) scheme, which includes stratospheric ozone chemistry processes, both gas-phase and heterogeneous, in a consistent embedded way for all locations using only the first four linear coefficients $c_i$ ($i = 0, 1, 2, 3$). Unlike previous stratospheric ozone linear schemes, the BMS scheme

is the first one to include heterogeneous chemistry effects and gas-phase chemistry in a consistent, implicit way in all terms of Eq.1. This new approach was shown to be very beneficial for the representation of ozone at high latitudes (Monge-Sanz et al., 2011).

In addition, the coefficients for the new scheme have been derived from a 3D full-chemistry model, in contrast to the default ozone scheme used by ECMWF which is based on a 2D photochemical model. This provides additional information on spatial

variability of the chemical tendencies represented by the coefficients. As Monge-Sanz et al. (2011) discussed, the new approach allows for more realistic interactions between parameterised ozone, radiation and temperature, and therefore better response and feedbacks to meteorological conditions, than previous ozone parameterisation approaches.

## 2.2 ECMWF model experiments

We have used the Integrated Forecast System (IFS) of ECMWF to run forecast experiments at medium-range (10 days) and

long-range (seasonal) time scales. The IFS configuration in each pair of experiments (control and new) differs only in the scheme used to model stratospheric ozone.

### 2.2.1 Medium-range experiments

A list of medium-range forecast experiments, corresponding periods and model configurations, is shown in Table 1. All model experiments, unless otherwise stated, are 10-day forecast runs using the 41r1 Cycle version of the IFS, which was operational

between May 2015 and March 2016. The model has been run at spectral resolutions of T511, T255 and T159, with 91 or 137 vertical levels, up to 0.01 hPa. Ozone concentrations are initialised from operational analyses at the start of each experiment (00UTC on day 1) and then, unless otherwise stated, ozone is left to evolve freely along the duration of the experiment using either the CD or the BMS scheme.

In the default IFS configuration the radiation scheme does not employ the prognostic ozone, instead it uses an ozone clima-

tology in the form of zonal-mean monthly-mean ozone values. In the IFS version used in this study, the ozone climatology is





derived from the MACC reanalysis (Inness et al., 2013), a dataset that covers the period 2003-2011 at T255 horizontal resolution on 60 vertical levels. However, for some of our experiments the standard IFS model has been adapted so that the prognostic ozone provides the input to the radiation scheme (Table 1), thus allowing for feedbacks between the ozone scheme and model dynamics.

### 2.2.2 Long-range experiments


Table 2 provides an overview of the long-range seasonal experiments we have performed for this study. We have performed two 5-member seasonal experiments, with May and November start dates and a 7-month integration range, for the period 2001-2010. One of these experiments is using the default ozone configuration (Cariolle scheme and no feedback onto radiation) and the other one is using the new BMS scheme interactive with radiation, both experiments with a T255 resolution and 137

vertical levels. Initial data come from ERA-Interim reanalysis fields (Dee et al., 2011). These two long-range experiments were performed with an experimental version of 41r1 that included research developments towards the new ECMWF seasonal forecasting System 5; the BMS stratospheric ozone model is one of such developments (Knight et al., 2016; Johnson et al., 2019). Another set of two seasonal experiments with the two ozone configurations has additionally been performed using the 41r1 coupled version of IFS operational in 2016, with an horizontal resolution T255 and 91 vertical levels. These are 3-member

ensemble experiments covering the 30-year period 1981-2010, to be able to compare results with the ERA-Interim reanalysis dataset.

### 2.3 Datasets for validation

In-situ observations from the global network of ozone sondes have been used for the validation of the model ozone vertical profiles. These observations come from the networks of the World Ozone and Ultraviolet Radiation Data Centre (WOUDC),

the Network for the Detection of Atmospheric Composition Change (NDACC), the Norwegian Institute for Air Research (NILU), the Southern Hemisphere Additional Ozonesondes (SHADOZ), the National Oceanic and Atmospheric Administration (NOAA). Additional measurements from the Coordinated Airborne Studies in the Tropics (CAST) and the Ozone Measuring Campaigns in the Arctic (MATCH) field campaigns have also been used. These ozone sonde observations provide a completely independent validation dataset as they are not assimilated by the (re)analyses used to provide the initial

conditions in our model experiments. Ozone observations from these ozone sondes are highly reliable ($\pm 5$ %) up to altitudes of 10.0-5.0 hPa. The global reanalysis produced by the Copernicus Atmosphere Monitoring Service (CAMS), CAMSiRA (Flemming et al., 2017), has provided additional comparisons for the ozone field in our model experiments. The ERA-Interim reanalysis (Dee et al., 2011) has been used to validate meteorological fields from our experiments.





## 3 Ozone distribution results

The distribution of stratospheric ozone and its time evolution for the experiments described above are shown in this Section for different latitudinal regions, atmospheric events and time scales. The impact of the new ozone scheme on meteorological fields at the different time scales will be discussed in Sect. 4.

### 3.1 Antarctic ozone hole

Figure 1 shows the time series of monthly averaged vertical profiles of ozone, for the period August 2012-February 2013, from
two experiments in which the ozone field is freely evolving along the 1-year period, but meteorology is initialised every 10 days (i.e. medium-range 10-day meteorological forecasts). One of the experiments uses the new BMS ozone, the other uses the default CD ozone scheme. By comparing ozone concentrations from these experiments with ozone sonde measurements (Fig. 1 left panel), we can clearly see the improvement obtained with the new ozone scheme over the Antarctic region, both in terms of vertical distribution of ozone concentrations and in terms of time evolution of the concentrations, especially during and after
the ozone hole season. The recovery of the ozone concentrations after October is also much more realistic with the new ozone model. The better performance of the new BMS scheme is mainly due to the new formulation of the heterogeneous chemistry treatment, that has been derived from realistic full-chemistry at all altitudes, unlike the $5^{th}$ term in Eq. 2 that is representative of an altitude of approximately 20 km. Moreover, the default scheme specifies a dependence on the square of the chlorine content $(Cl_{EQ})^2$ which is not representative of typical atmospheric conditions in the activated polar vortex (e.g. Searle et al., 1998).

#### 3.1.1 Antarctic vortex split 2002

In 2002 a vortex split was observed for the first time over the Antarctic region, following a stratospheric major warming in the winter stratosphere (e.g. Krüger et al., 2005; Roscoe et al., 2005). Such events are relatively common over the Arctic, where a sudden stratospheric warming can take place in almost half of winters (e.g. Butler et al., 2017). The unusual Antarctic event in 2002 had an enormous impact on the evolution of the ozone hole, which was one of the smallest ones ever recorded. The
atypical characterstics of this 2002 vortex split are a challenging benchmark to evaluate the adaptability of the new ozone scheme to meteorological and chemical conditions that very significantly differ from climatological conditions in this region. Figure 2 shows the time evolution of the $O_3$ vertical distribution at two Antarctic radiosonde stations, Syowa (69°S, 40°E) and the South Pole (90°S). Results are from 10-day forecast experiments in which the ozone field is left to evolve freely along the whole length of the experiment ($1^{st}$ August 2002-$1^{st}$ January 2003).
After the vortex split at the end of September, the ozone destruction over the station of Syowa stopped and ozone concentrations quickly recovered; this is well captured by the new scheme but not by the default scheme which shows too low maximum concentrations compared to observations for October-December. Over the South Pole station in October both model schemes simulate larger concentrations than observed but in November-December it is again the new scheme that simulates a more realistic recovery of ozone values. The more realistic link with temperature in the new scheme, for gas-phase and heterogenous
chemistry, makes it capable to respond to the rapid changes in atmospheric dynamics that take place in an event like the 2002





Antarctic vortex split; Monge-Sanz et al. (2011) showed that this kind of response resembles that of a full-chemistry 3D global model. In the default scheme, the artificially detached heterogenous term cannot adapt in the same way.

### 3.1.2 Interannual variability of the ozone hole season

For seasonal timescales, several experiments were carried out in the framework of the EU FP7 project SPECS (Seasonal-
to-decadal climate Prediction for the improvement of European Climate Services), see Knight et al. (2016). An experimental version of the ECMWF seasonal system SEAS5 (Johnson et al., 2019) was used here to perform 5-member seasonal forecast experiments with starting dates in May and November, covering the period 2001-2010. They use a T255 horizontal resolution and 137 vertical levels from the surface up to 0.01 hPa, and initial conditions from ERA-Interim reanalysis fields.

Figure 3 shows the time evolution of the stratospheric ozone hole area and depth for these two model experiments, and
for the CAMSiRA Reanalysis ozone field for the overlapping period (2003-2010). To quantify the intensity and extent of the ozone hole, Fig. 3 displays the total ozone column (TOC) minimum value and the ozone hole size fraction (area with TOC values below 220 DU for latitudes over $62°$S) from July to December. Both model experiments capture the formation and evolution of the ozone hole, but the new ozone scheme produces more ozone loss, and over a larger area, than that reproduced by the CAMSiRA reanalyses. Total ozone column values from CAMSiRA are known to be too large (by up to 30 DU) over
the Antarctic region when compared to independent observations (Fig. 16 in Flemming et al., 2017) for the period considered here; this, together with the results we have shown above in Sect. 3.1, adds confidence to the ozone hole structure obtained with the new BMS scheme in these seasonal experiments.

Interannual variability is more realistic with the new BMS scheme compared to the reanalysis, while the default scheme does not show enough variability, especially for the fraction area covered by the ozone hole, which manifests that the description
of heterogenous chemistry processes in the default scheme is not realistic enough to deal with meteorological interannual variability in the Antarctic region. Since CAMSiRA uses the CD scheme in the stratosphere, Fig. 3 also suggests that with the new BMS scheme, assimilation increments in the reanalysis would be reduced for the ozone field.

The formation of the hole commences earlier when using the new ozone scheme, this is at least partly due to the fact that the new ozone scheme is able to capture the ozone loss that in late winter starts to occur at the edge of the polar vortex, while
the heterogeneous treatment in the scheme currently used by ECMWF cannot reproduce this process. The ozone hole closure shows large interannual variability in the reanalysis and the simulation with the new ozone scheme is in better agreement regarding this interannual variability, although the model simulations shown in Fig. 3 end in late November and years with closure dates beyond that cannot be compared. For those years with an early closure, the overall duration of the ozone hole is similar with the new scheme and in the reanalysis, but the timing of the ozone hole formation occurs earlier in the new ozone
model run than in the reanalysis. In contrast, the duration and extent of the ozone hole with the default ozone scheme is reduced compared to both the reanalysis and the new scheme.

It is also worth noting that these ozone hole diagnostics are related to the total ozone column (TOC), and that realistic TOC values do not necessarily imply that the depletion in the model is occurring at the right altitudes. For similar TOC values over the Antarctic, we have shown that the new BMS scheme provides a much more realistic ozone vertical profile than the





default ozone scheme in the ECMWF model (Sect. 3.1), also in agreement with previous studies using the BMS scheme
(Monge-Sanz et al., 2011; Jeong et al., 2016; Badia et al., 2017).

   Regarding the duration and extent of the ozone hole, several studies have shown how the Antarctic stratospheric ozone hole
feeds back to dynamics and radiation causing changes in tropospheric winds and climate (e.g. Kang et al., 2011; Polvani et al.,
2011; Orr et al., 2012; Haase et al., 2020). It is therefore important that future Earth System Models use a stratospheric ozone
description that realistically captures the ozone hole intensity and evolution, to correctly simulate tropospheric trends.

### 3.2   Tropics and Midlatitudes

#### 3.2.1   Tropics

Over the tropics both schemes provide realistic ozone distributions in our model runs, although the maximum values are
underestimated by both linear schemes compared to ozonesondes (figure not shown). This negative bias over the tropics was
already documented by Monge-Sanz et al. (2011), their study showed that the upper limit imposed by the ozone climatology
term in Eq. 1 means that a linear scheme of these characteristics cannot provide higher concentrations over this region than
the values provided by the climatology, which in the case of biases being present in the climatology term is a caveat for these
linear ozone schemes over certain regions. Monge-Sanz et al. (2011) showed that a different choice of ozone climatology term
can improve this tropical bias, although caution must be taken on selecting the climatology to ensure that performance over
other regions is not degraded. This showed that the use of a carefully chosen climatology term, either from an updated run of
the parent full-chemistry CTM or from a recent observation-based climatology, can improve this bias over the tropics.

#### 3.2.2   Midlatitudes

One of the longest ozone records in Europe corresponds to the Alpine station of Hohenpeissenberg (Fig. 4). At this midlatitude
location (47°N, 11°E) the two linear ozone schemes capture well the overall annual cycle (low ozone concentrations in autumn-
winter and high ozone concentrations in spring-summer). However, both schemes behave differently in terms of biases. From
the integrated profiles and the corresponding biases compared to observations (low panels in Fig. 4), we can see that the
new scheme produces ozone column values in close agreement with observations from August to December 2012, and then
underestimates column values (up to -0.7 mPa) from January to July 2013; the default ozone sheme however overestimates
column values all along the year by up to 1.0 mPa. This overall behaviour is representative of most European midlatitude
stations (figure not shown). The overestimation with the default ozone may partly be caused by ozone concentrations at high
latitudes being larger, and then air richer in ozone being transported towards midlatitudes.

### 3.3   Arctic ozone

Ozone loss in the Arctic region does not usually reach the levels found in the Antarctic. However, ozone depletion over
the Arctic presents high interannual variability because of the large variations in this region winter dynamics. Arctic winter
temperature variations and vortex variability affect the formation of polar stratospheric clouds (PSCs), and thus the chemical



processing that leads to ozone destruction can be largely different from year to year. The occurrence of sudden stratospheric warming (SSW) events, with temperatures in the polar stratosphere experiencing very rapid increases, lead to significantly lower Arctic ozone loss than during cold Arctic years without vortex disturbances (e.g. Monge-Sanz et al., 2011; Strahan et al., 2016).

Ozone loss in the Arctic region is largely dependent on the number of consecutive cold days during winter, and therefore on dynamics and the occurrence of SSWs. The Arctic winter 2015/2016 saw anomalous polar vortex behaviour. Polar stratospheric vortex winds were stronger than usual since early winter, then in mid-winter a minor SSW took place, followed by a major warming in late winter (e.g. Manney and Lawrence, 2016). During the minor warming in February 2016, PSCs were seen over North England mid-latitude locations; the closest ozonesonde station is the Irish station of Valentia (52°N-10°W).

To take into account feedbacks between ozone and dynamics during this event we ran two experiments, one with the new BMS ozone scheme interactive with radiation and one using the default ozone configuration in which the radiation sees a climatological ozone field. Monthly averaged profiles of differences (model - observations) at Valentia station in January and February 2016 show that both experiments underestimate the maximum of the ozone profile, but differences between forecast ozone and observations are smaller when using the new BMS ozone scheme interactive with radiation than using the default
ozone configuration, at all pressure levels except around 150 hPa (Fig. 5).

Similar differences are also seen for Arctic stations. Arctic ozone profiles corresponding to day 10 in the forecast runs are shown in Fig. 6 together with the corresponding ozonesonde observations for February 2016. Although both model runs overestimate the concentration values, and underestimate the altitude, of the ozone maximum compared to the ozonesonde profiles, the model run with the new BMS scheme results in concentrations closer to observations; the percentage differences
(forecast - observed) are up to 10 % smaller with the new ozone, except at 250 hPa.

SSW events like this one in winter 2015/2016 are good case studies to assess the new stratospheric ozone scheme. Important feedbacks between ozone and dynamics take place in this type of events: The occurrence of a SSW reduces the amount of ozone loss over the Arctic, therefore more ozone is available to absorb UV radiation, which contributes to a further increase of temperature in the region. The fact that past versions of the operational ECMWF forecast model reproduced SSW events overall
weaker (colder) than observed (e.g. Diamantakis, 2014) is a further indication that a forecast model cannot properly account for these feedbacks when using an ozone climatology in the radiation scheme. A realistic scheme for prognostic stratospheric ozone is therefore able to contribute to a better reproduction of SSW events and their feedbacks within the model.

## 4  Impact on meteorological fields

The previous sections have shown the improved stratospheric ozone distribution and variability obtained when using the new
BMS ozone model in the ECMWF system. From a weather and climate modelling perspective, we are interested in how the new representation of stratospheric ozone affects meteorological fields. To evaluate this, the prognostic ozone scheme has been made interactive with the radiation scheme in the ECMWF model. The corresponding impact on meteorological variables





has been compared with results from the default ECMWF ozone configuration in which the radiation scheme uses an ozone climatology.

## 4.1 Impacts on medium-range forecasts

The mean error in the temperature field is shown in Fig. 7 for two forecast model experiments in which the prognostic ozone has been made interactive with the radiation scheme. Both experiments are 10-day forecast covering the period August 2012-July 2013. The only difference between the configuration of the two experiments is in the prognostic ozone model: one of them uses the CD ozone, the other one the new BMS ozone. In the stratosphere (above 100 hPa), the new ozone clearly reduces the model temperature bias by up to 1 K. Smaller improvements can also be seen for lower levels in the extratropics (Fig. 7). The tropical region at 100 hPa is the only region in which the use of the new ozone scheme increases the temperature mean error. This is most probably attributable to the negative bias in tropical ozone exhibited by the BMS scheme. Although the negative ozone bias is found above 100 hPa, it means that more UV can reach lower altitudes (less ozone above), therefore less ozone below (more dissociation by UV); and with less ozone there is also a decrease in local T, leading to the larger negative T bias over the tropics at 100 hPa shown in Fig. 7. For tropospheric levels there is an overall improvement in the T mean error, although not everywhere for all leadtimes, but results for the troposphere are not statistically significant.

To display the statistical significance of these results, the differences in normalised error change in the temperature field, and corresponding significance 95% bars, are shown in Fig. 8. The improvement in the model error in the stratosphere above 100 hPa is clearly evident and statistically significant with the new ozone. For altitudes below 100 hPa, it shows small increases in model error with the new scheme but these are not statistically significant, except in the tropical UTLS. Interpreting results in the tropical UTLS is not straightforward due to the many interplaying factors in this region, but the negative concentrations bias exhibited at higher altitudes over the tropics is most probably playing a role. In summary, the new scheme, therefore, improves the temperature behaviour in the stratosphere without degrading the temperature field in the troposphere; the only atmospheric region where there is some degradation in the T field is the tropical LS and this is at least partly due to a known bias in the new ozone scheme.

A similar comparison can be performed between the forecast experiment using the prognostic BMS ozone interactive with radiation and the default ECMWF operational configuration in which the radiation scheme uses an ozone climatology. Fig. 9 shows the mean error in temperature for these two experiments and Fig. 10 shows the corresponding differences in normalised error change with the 95% significance bars. These results show a similar pattern to those from the comparison of the two prognostic schemes, i.e. the BMS scheme provides an improved temperature field in the stratosphere compared to the default climatology, with the exception of the tropical region at 50 hPa and the NH extratropics at the same 50 hPa level. The new ozone also provides mean error improvements for the troposphere outside the tropics although results are only marginally statistically significant or not significant. Note also that differences in the troposphere are one order of magnitude smaller than in the stratosphere. In the tropical troposphere a small degradation can be found, which unlike in the comparison of both schemes now becomes statistically significant. This is related to the fact that this type of ozone linear models is not designed for tropospheric use, and in the troposphere the use of a realistic climatology could be considered a good alternative for NWP





purposes.

The rest of this paper focuses on the evaluation of meteorological impacts for experiments using the new BMS prognostic
ozone compared to the control experiments using the default climatology configuration. It is worth noting that the ECMWF
operational model has gone through version updates after our study, and the ozone climatology has been updated follow-
ing Hogan et al. (2017). Although beyond the scope of this paper, comparison of biases for the currently operational default
configuration should be a matter of future investigation.

## 4.2   Impacts during SSW events

The winter stratospheric variability in the NH high latitudes is dominated by the occurrence of sudden stratospheric warmings
(SSWs); since these SSW events can lead to surface cold outbreaks over NH midlatitudes (e.g. Baldwin and Dunkerton, 2001;
Kolstad et al., 2010; Lehtonen and Karpechko, 2016), it is important for the forecast model of an NWP system to simulate
them as realistically as possible. During the SSW that took place in early February 2016, temperature anomalies reached max-
imum values on the $7^{th}$ February 2016 between 5 and 20 hPa (e.g. Manney and Lawrence, 2016). Two medium-range forecast
experiments are used to assess the role of the new ozone model in this SSW event, one using the new BMS ozone interac-
tive with radiation and one using the default model configuration in which the radiation scheme sees an ozone climatology.
These experiments cover the period $15^{th}$ December 2015 - $14^{th}$ February 2016; to compare them we are showing some of the
diagnostics used by Diamantakis (2014).

Figure 11 shows that, with the new prognostic ozone scheme, temperature at 5 hPa becomes warmer over the Eastern Arctic
region (up to 20 K warmer) compared to the default model configuration (Fig. 11a,b,d), bringing it closer to the operational
analysis (Fig. 11c). We have compared t+240 h in the forecast experiments, from the $28^{th}$ of January forecast, against the
corresponding operational analysis for the $7^{th}$ February, to allow for the maximum ozone response. The improvement seen at
5 hPa is also seen at lower altitude levels (figure not shown).

For these two experiments we have also examined the impact on the wind field. Figure 12 shows the change in RMS error
for the meridional wind velocity for different lead times, averaged over the experiment duration. The improvement in wind
errors when BMS ozone is used consistently increases with lead time, and is transferred from the stratosphere down to the
troposphere for high latitudes from day 6. By day 10 in the forecast the error reduction in the wind field is statistically significant
in the troposphere. During SSW events, when both temperature and ozone distributions change rapidly in the stratosphere, a
climatology-based ozone field cannot pass information to the radiation code that resembles the actual atmospheric situation,
therefore the model misses the potential source of tropospheric predictability that comes from the downward propagation of
the stratospheric signal.

## 4.3   Long-range impacts

We have performed two seasonal experiments with start dates in May and November, covering the period 1981-2010, using
an horizontal resolution of T255, 91 vertical levels and three members. The control experiment uses the MACC ozone cli-





matology inside the radiation code, while the BMS experiment uses the new prognostic ozone interactive with the radiation code. Temperature differences with respect to ERA-Interim for these two experiments are shown in Figure 13, averaged over DJF (upper panels) and MAM (lower panels). There is a clear improvement around 50 hPa when using the new ozone scheme for both seasons and all latitudes, especially over the SH mid and high latitudes. In these SH regions the BMS prognostic ozone reduces differences by up to 4.0 K. Also for levels above 20 hPa differences with respect to ERA-Interim are reduced,

especially in the summer hemisphere, by more than 1.0 K during DJF and MAM. These results have shown that a prognostic ozone field contributes to more realistic temperatures in the SH stratosphere than a climatology, also for the seasons following the Antarctic ozone hole months.

Figure 14 shows the zonal averaged differences in zonal wind between the two experiments for the SON season. The new ozone experiment shows stronger zonal winds over the Antarctic vortex edge latitudes between 20-400 hPa, which is physically

linked to the lower concentrations of ozone simulated by the new scheme over this region compared to the default climatology. This strengthening of winds is in overall agreement with the findings in Son et al. (2008). Seasonal experiments performed under the SPECS[1] EU project also showed improvements in the equatorial winds and the QBO signal when the new BMS prognostic ozone was made interactive with radiation (Knight et al., 2016).

### 4.3.1 North Atlantic Sector

Moving to the North Atlantic Sector winter, we find that the new ozone scheme has a positive impact on the signal to noise ratio; accordingly, using ERA-Interim as reference, the correlation value for the winter North Atlantic Oscillation (NAO) index is almost doubled compared to the default configuration (Fig.15), increasing from 0.25 in the control experiment (default ozone climatology) to 0.44 in the experiment with the new BMS prognostic ozone. These experiments, where the only difference is the stratospheric ozone representation, allow us to attribute this increase in NAO model performance to stratospheric sources.

A more realistic stratospheric ozone distribution improves the ozone concentration gradients between the Pole and the equator, which modifies the latitudinal heating gradient in the LS region compared to the default model configuration. This affects LS wave breaking and winds and has also an effect on the altitude distribution of the tropopause. The combination of these effects impacts the tropospheric pressure gradient between low and high latitudes and therefore the NAO signal.

These results show that a more realistic stratospheric ozone field contributes to a more realistic stratosphere-troposphere cou-

pling in the model. The links between the NAO and winter time weather over Europe are well established (e.g. Cattiaux et al., 2010; Buehler et al., 2011; Scaife et al., 2014) therefore, using a more realistic stratospheric ozone description increases the potential to exploit stratospheric sources for improved tropospheric weather prediction in the Atlantic Sector.

### 4.3.2 Antarctic polar vortex

In the Southern Hemisphere the seasonality and interannual variability of the polar vortex and the ozone layer are closely re-

lated. To investigate this last part of our study we have had access to the new seasonal system of ECMWF, SEAS5 (Johnson et al., 2019). Two SEAS5 seasonal experiments are compared here, one seasonal experiment uses SEAS5 with its default configu-

[1]Seasonal-to-decadal climate Prediction for the improvement of European Climate Services





ration (ozone climatology in the radiation scheme), the second SEAS5 experiment uses the same configuration except that ozone is replaced by the BMS prognostic ozone model interactive with radiation. The interannual variability of the SH polar vortex is shown in Fig. 16 for these seasonal experiments initialised on the $1^{st}$ of August over the period 1993-2015 (2002

has been excluded from the analysis shown in this figure). From the top panel in Fig. 16 it can be seen that the seasonality of the stratosphere in this region is not realistic with the default SEAS5 compared to ERA-Interim; the vortex shift-down occurs too early compared to the reanalysis. When the new BMS prognostic ozone is used the timing of the SH polar vortex is in much better agreement with ERA-Interim, and the interannual variability increases. Byrne and Shepherd (2018) in their study using ERA-Interim reanalysis, found that Antarctic ozone depletion has caused a seasonal delay in the breakdown of the SH

polar vortex along the period 1980-present, and they also pointed out that feedbacks between ozone and dynamics may be responsible for the increase in the interannual variability observed in the SH polar vortex. Our results with the SEAS5 experiments confirm their findings, showing that a realistic interactive prognostic ozone field is needed to reproduce the SH polar vortex behaviour, while an ozone climatology does not provide enough information for the model to reproduce all necessary feedbacks with dynamics.

Figure 17 shows the correlation of the polar-cap averaged geopotential height in the SH, between the seasonal experiments used in Fig. 16 and ERA-Interim. For the period 1993-2015, excluding year 2002, the correlation patterns are similar for the standard SEAS5 and SEAS5 with BMS ozone, although the new ozone contributes to significantly larger correlation values for levels above 100 hPa in November-January. This is consistent with the more realistic timing in the polar vortex simulated with the new BMS ozone. If year 2002 is included in the analysis (lower panels in Fig. 17), the difference is, as expected, more

evident. With the new ozone the correlation clearly increases during September-October-November (SON), also at tropospheric levels, where the deafult SEAS5 simulation was not showing any significant correlation. This can be explained by the ability of the new ozone model to realistically respond and feedback to the rapid dynamical changes that took place during the unusual Antarctic vortex split of year 2002; in Section 3.1.1 we have shown the capability of the new BMS scheme to simulate realistic ozone distributions during the 2002 Antarctic vortex split. Results in Fig. 17 demonstrate that using a stratospheric ozone

model capable of reproducing realistic evolution of ozone vertical concentrations also improves meteorological fields, both on average over the whole period considered, 1993-2015, and also during very unusual meteorological events such as the 2002 Antarctic vortex split. A model without a realistic description of stratospheric ozone understimates the role of the stratosphere in shaping tropospheric meteorological fields at different timescales.

## 5   Discussion and Conclusions

In this section we summarise the main findings of our study, and discuss further work plans and recommendations deriving from our investigations.





## 5.1 Summary

We have implemented the stratospheric ozone model by Monge-Sanz et al. (2011) (also known as BMS model) in the ECMWF system, compared its performance to that of the default ozone used by ECMWF, and assessed its impacts on meterological fields at medium-range and seasonal time scales. The BMS scheme is the first stratospheric ozone linear model that consistently accounts for heterogeneous chemistry (e.g. ozone destruction due to polar stratospheric clouds), instead of using a separate adhoc term, providing a more realistic link with temperature and radiation. The new approach is in better agreement with the current scientific knowledge of chemical and physical processes that affect stratospheric ozone (WMO, 2019) than approaches adopted by previous linear ozone models (McLinden et al., 2000; McCormack et al., 2006; Cariolle and Teyssèdre, 2007).

The present study is, to the best of our knowledge, the first time that the impacts of a stratospheric ozone model are assessed at different NWP timescales to evaluate its performance towards its implementation in a seamless model system. We have shown that the new scheme provides significantly better ozone distribution and variability in the ECMWF model than the currently default ozone configuration, showing particularly good agreement with observations over the high southern latitudes and the ozone hole season, even during the unusual atmospheric conditions of the 2002 Antarctic vortex split.

When used interactively with radiation in the ECMWF model, the BMS ozone scheme reduces stratospheric temperature biases both in medium-range and seasonal time scales, compared to the default ECMWF model configuration in which the radiation scheme uses an ozone climatology, and improves temperature and wind fields during Arctic SSWs. We have also shown that the BMS ozone improves the NAO signal in seasonal model runs, therefore contributing to a more realistic stratosphere-troposphere coupling in the model. The interannual variability and seasonality of the SH polar vortex is also improved when using the BMS ozone model in runs performed with the ECMWF seasonal system (SEAS5), compared to the default SEAS5 configuration which uses an ozone climatology. All this demonstrates that the BMS scheme is realistically linked to temperature and dynamics and therefore well prepared to adapt and feedback to rapid changes in meteorology, the same adaptability cannot be achieved with an ozone climatology in the radiation scheme.

Our results also provide evidence for the need of a realistic prognostic stratospheric ozone field in ESMs for these models to perform more accurately at different time scales. The realistic stratospheric concentration values obtained with the BMS scheme, its high adaptability to usual and unusual meteorological conditions at different time scales, together with its low computational cost, make the BMS scheme an excellent option to model stratospheric ozone within ESMs. The BMS scheme is able to model stratospheric ozone with a degree of complexity that provides realistic stratospheric ozone distributions, of comparable quality to the ozone field from world-leading full-chemistry models, while keeping low computational costs suitable for resolution and production times required for weather forecasting (both at medium-range and seasonal time scales). Future work should also investigate the performance of the BMS scheme within other global models and exploit its benefits for different ESMs applications beyond medium-range NWP and seasonal prediction, into climate time scales.

Next, we briefly discuss ongoing further developments of the BMS model version we have used here, and how they are expected to improve the results obtained with the current version. We also provide a summary of benefits that the implementation





of a realistic prognostic stratospheric ozone brings to seamless Earth System Models, from medium-range NWP forecasts to
seasonal prediction and Reanalysis production.

## 5.2    Ongoing and Future work

To tackle the tropical bias exhibited in model runs with the BMS scheme, Monge-Sanz et al. (2011) tested the use of a different
climatology reference based on observations, and although results improved over the tropics (Fig. 7 in their paper), concentra-
tions over high latitudes were degraded by the different climatology. More recent versions of the parent full-chemistry CTM
have solved the tropical bias issue and can now be used to derive a new version of the climatology terms that would not suffer
from this tropical negative bias. To reduce differences with observations at high latitudes the scheme could use coefficients that
are provided twice a month, instead of just on the 15th of each month, which allow to simulate more realistic loss rates for the
sunlight levels found in the polar early spring. As a further method to improve the simulation of ozone distributions, unlike the
default CD scheme, the BMS scheme coefficients could be provided in a 3D grid, instead of 2D averaged values. Following
the method by Monge-Sanz et al. (2011) a newer version of the parent CTM can be used to compute a more advanced version
of the BMS ozone scheme. This newly derived version can be obtained at higher horizontal and vertical resolution, and the
tendencies computed with CTM runs driven by more accurate meteorology, e.g. from ERA-5 reanalyses fields. Solar effects
are now also part of the latest CTM version, thus solar variability effects can also be incorporated in the linear ozone model
by deriving tendencies corresponding to solar maxima and solar minima conditions. The production of this new version of the
BMS scheme is now part of ongoing work. As additional future developments the ozone scheme can also include higher order
terms to take into account second-order nonlinear effects, as well as links to other main stratospheric species like CFCs and
$CH_4$ Monge-Sanz et al. (2013).

### 5.2.1    Tropospheric ozone treatment

The ozone model discussed in this study, as other ozone linear schemes like the default one used by the ECMWF system, are
designed for the stratosphere; their use is not recommended for the tropospheric region, where ozone is affected by highly non-
linear processes involving pollutants and ozone precursors. A realistic representation of tropospheric ozone is the full-chemistry
approach, but this is still unviable for operational high resolution models due to high computational costs. Alternatives for rep-
resenting tropospheric ozone include the use of an up-to-date climatology based on observations or reanalysis that would be
merged to the prognostic ozone in the stratosphere.

### 5.2.2    Benefits for seamless Earth System Models

A realistic stratosphere is increasingly recognised as one of the keys to develop seamless Earth System Models, due to the
role it plays for tropospheric processes at different time scales, from weather to climate. The ozone model in our study is a
valuable contribution to achieve seamless use of emerging ESMs, as it offers similar accuracy to a full-chemistry model for
stratospheric ozone, allowing for ozone-climate feedbacks that level with those provided by current chemistry-climate models



(CCMs) with interactive stratospheric ozone, while keeping the computational cost affordable for weather forecast applications and resolutions.

Additionally, by implementing a realistic scheme for prognostic ozone, and ideally also for other radiative active gases in the stratosphere (see e.g. Monge-Sanz et al., 2013), the system will be better prepared for the eventual operational use of interactive
full-chemistry, as several feedbacks within the model will already have been investigated with the ozone model we propose, allowing for compensation errors to be identified and possibly eliminated.

### 5.2.3 Benefits for long Reanalyses

Long reanalyses have become an essential part of weather and climate scientific research and applications. Recent major international projects, like the SPARC[2] Reanalysis Intercomparison Project (SRIP), part of the World Climate Research Program
(WCRP) core activities, have identified areas that will need more attention in the production of future reanalyses in order to represent a more realistic stratosphere (Fujiwara et al., 2017). The representation of stratospheric ozone in the atmospheric models used to produce the reanalyses is one of these aspects (Monge-Sanz et al., under review, 2020). The adaptability the new BMS stratospheric ozone scheme shows to very different meteorological situations makes it an excellent candidate for use in oncoming Earth System Reanalysis, where climate forcings and their feedbacks will need to be accurately accounted for.

*Author contributions.* BMMS designed the study, implemented the new ozone scheme in the ECMWF system, performed experiments, results validation and interpretation, wrote the paper and coordinated co-authors' contributions. AB, NB, MD, JF, LM, IP and AW contributed to data analysis and results interpretation. LM and IP also contributed to performance of seasonal experiments. LJ designed the visualization tool used for validation against ozone profile observations. LJG, RJH, NW, MPC and TGS provided insightful feedback and contributed to review draft versions of the manuscript.

*Competing interests.* The authors declare that they have no conflict of interest.

*Acknowledgements.* This study was partially funded by the MACC-II and SPECS FP7 EU projects. BMMS and LJG also acknowledge funding from the UK Natural Environment Research Council (NERC) through the ACSIS project (North Atlantic Climate System Integrated Study) led by the National Centre for Atmospheric Science (NCAS). The first author is very grateful to Adrian Simmons, Agathe Untch and Jean-Jacques Morcrette for many helpful discussions and their valuable initial guidance with the ECMWF system; special thanks also to
Franco Molteni for useful discussions during the preparation of this manuscript. We also thank Paul Burton, Gabor Radnoti and the ECMWF User Support Team for their help with the IFS technical environment.

---

[2]Stratosphere-troposphere Processes and their Role in Climate





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





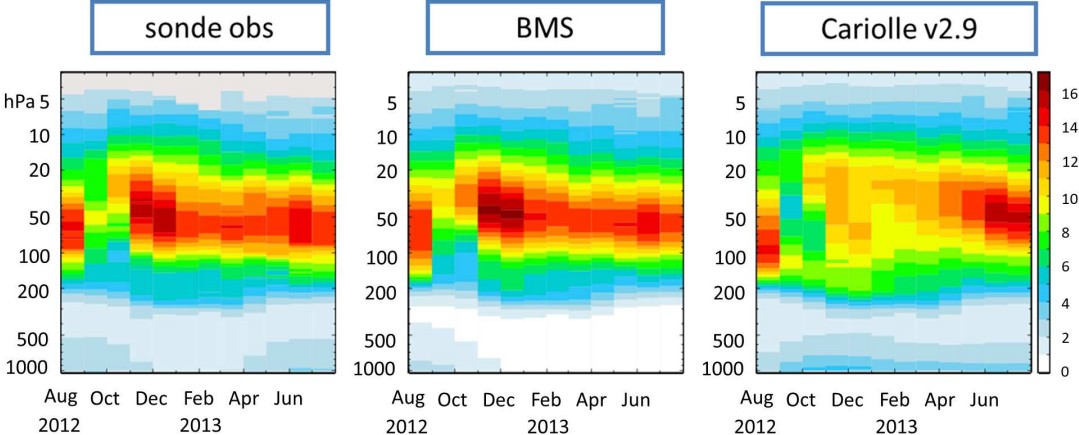

**Figure 1.** Monthly averaged time series of the vertical profiles of ozone (mPa) over southern high latitudes (55°S-90°S) for the period August 2012-August 2013. Panels show ozone concentrations as measured by ozone sondes (left), as simulated with the new ozone scheme in IFS (middle) and as simulated with the default IFS scheme (right). The vertical distribution and the time evolution of the ozone hole and its recovery are better represented by the new scheme.

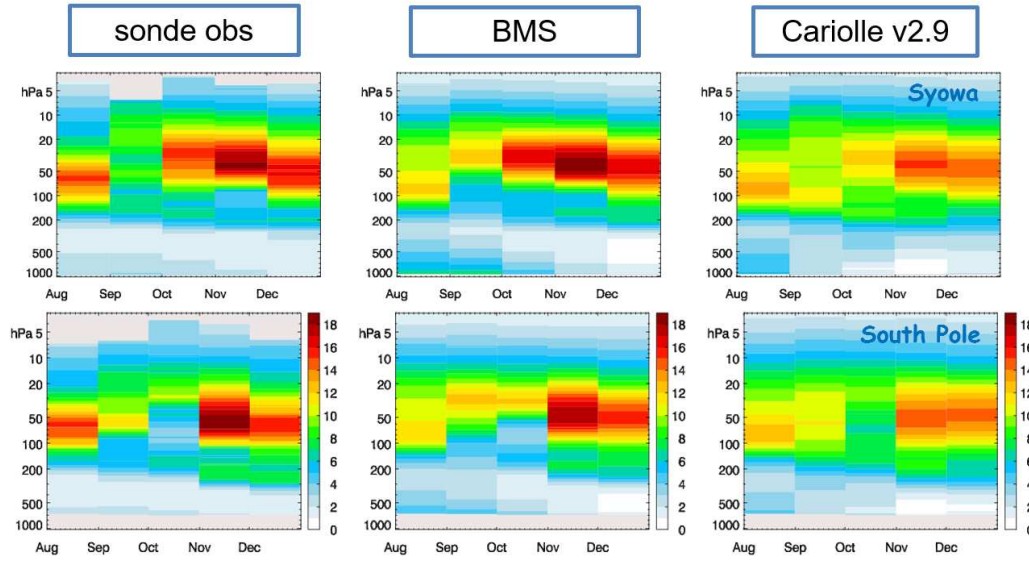

**Figure 2.** Monthly averaged time series of vertical profiles of ozone (mPa) over the Southern Hemisphere stations of Syowa (69°S, 40°E), upper panels, and the South Pole (90°S), bottom panels. Period shown is August 2002-December 2002, corresponding to the first Antarctic vortex split and ozone hole recovery afterwards.

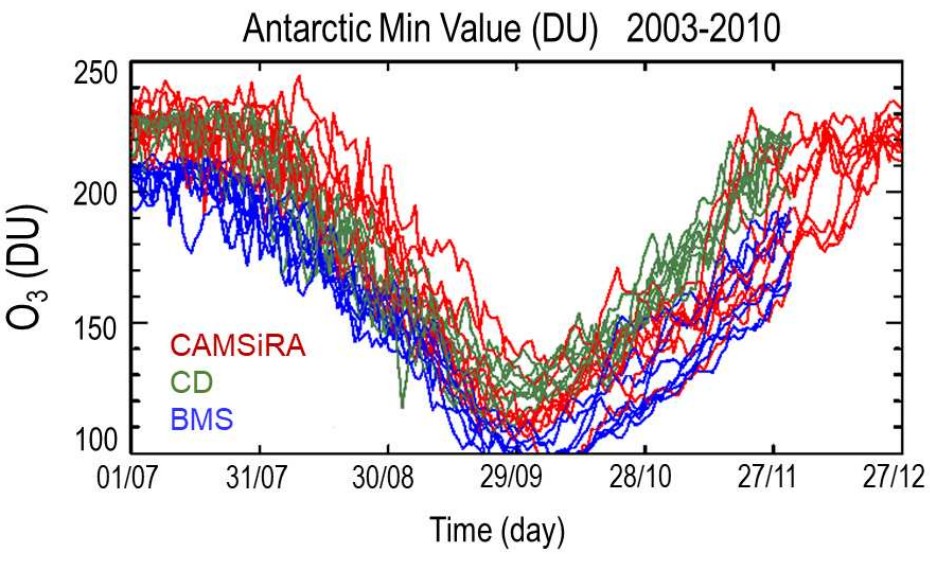

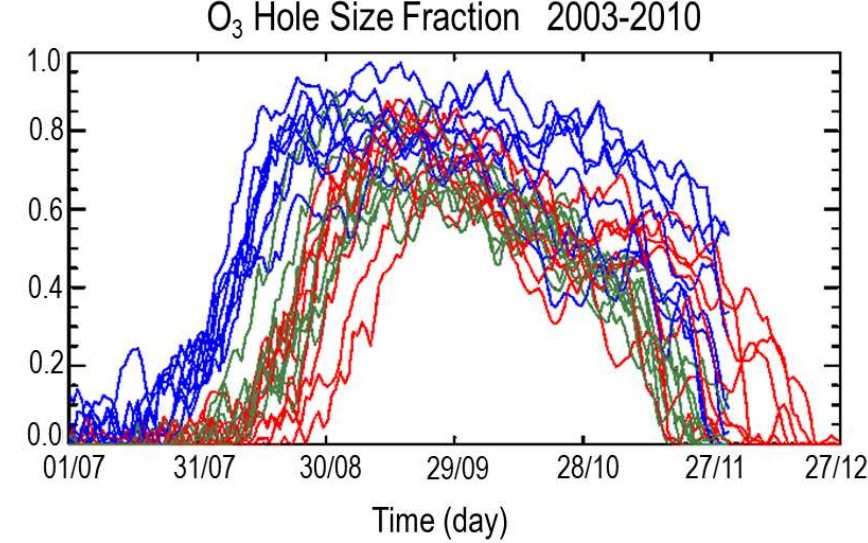

**Figure 3.** Time series of daily values of the total ozone column (DU) over the Antarctic region (top panel) and of the ozone hole size fraction (bottom panel) from July to December, for the years 2003-2010. Model results come from long range model ensemble experiments with May starting date (see text for details) using the default stratospheric ozone scheme (green lines) and the new stratospheric ozone scheme (blue lines). The values corresponding to the CAMSiRA reanalyses are also shown (red lines).

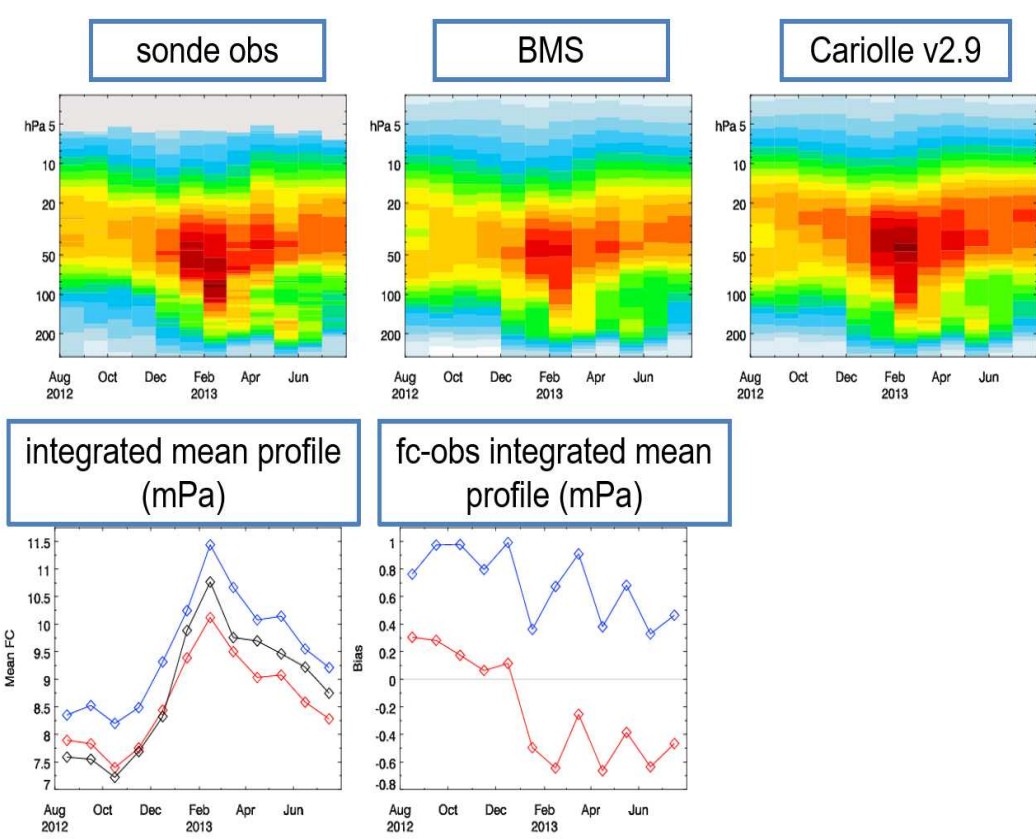

**Figure 4.** Monthly averaged time series of the vertical stratospheric profiles of ozone (mPa), for the period August 2012 - July 2013, over the European Alpine Hohenpeissenberg station (47°N,11°E), from the observed profiles (left), the model experiment with the new ozone scheme (middle), and a model experiment using the default ozone (right). The values for the integrated mean vertical profiles are shown at the bottom left for observations (black), model experiment using default option for ozone (blue) and model experiment using the new ozone scheme (red). The bottom right panel shows the corresponding differences in the integrated mean vertical profile between the two forecast (fc) runs and the observations (obs), for the experiment using default option for ozone (blue) and the experiment using the new ozone scheme (red).

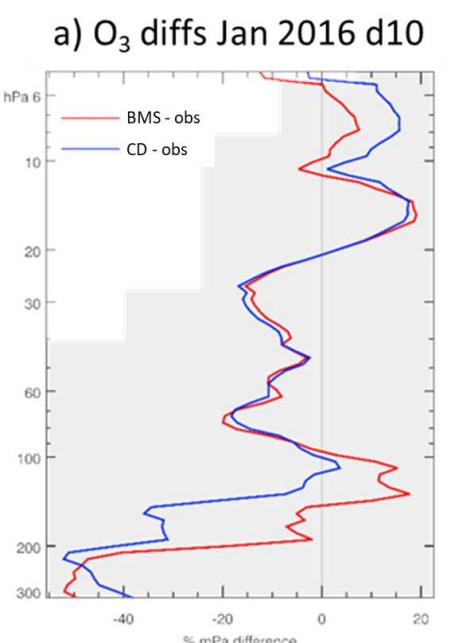
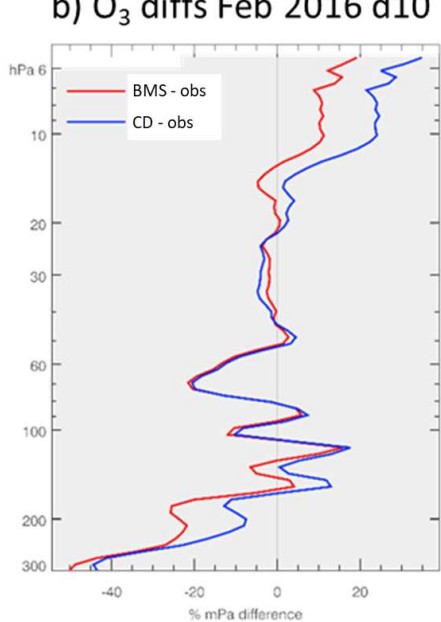

**Figure 5.** Averaged vertical profiles of ozone percentage differences for the Irish station of Valentia (52°N-10°W) in a) January and b) February 2016 for the model run using the new ozone scheme interactive with radiation (red) and the default ozone configuration (blue). Profiles used, both for model runs and observations, correspond to day 10 in the forecast runs.



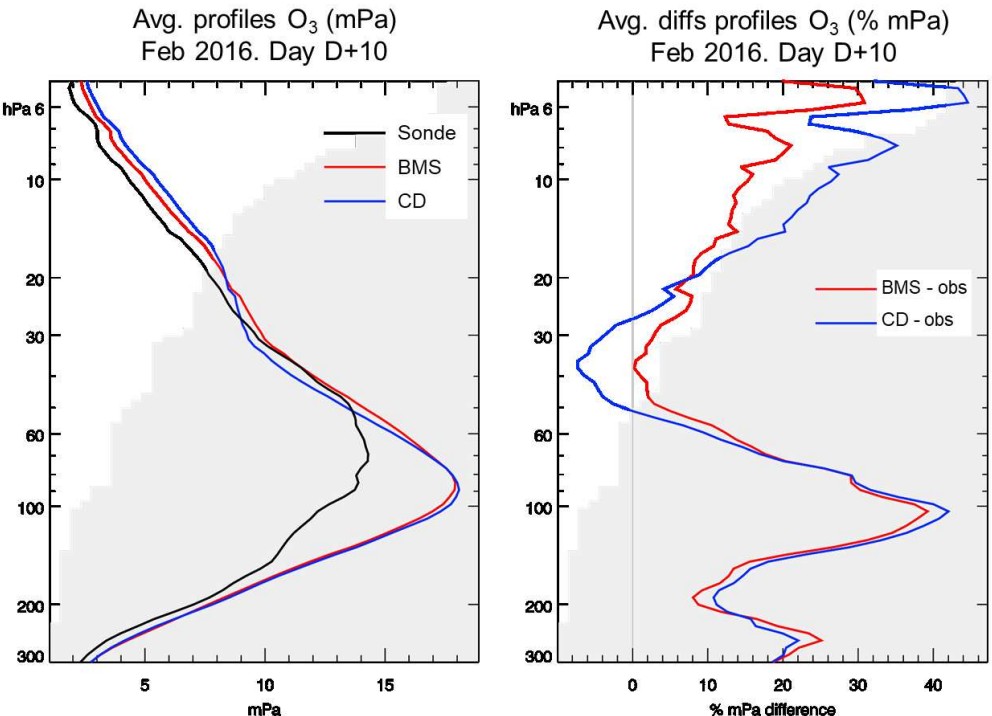

**Figure 6.** Averaged vertical profiles of ozone (mPa) over the Arctic region in February 2016 (left panel) from ozonesonde observations (black), from a model run using the new BMS ozone scheme (red) and a model run using the default ozone configuration (blue). The model profiles are for day 10 in the forecast run. The percentage differences in ozone concentrations between the forecast runs and the observations are also shown (right panel).

**Figure 7.** Mean error in the temperature field (K) compared to the operational ECMWF analyses, as a function of leadtime for the forecast experiment using the new BMS ozone (black) and the one using the CD ozone (red), both with prognostic ozone interactive with radiation. The two model runs consist of 10-day forecasts covering 1-year period, from August 2012 to July 2013. The figure shows three latitudinal bands, 90°S-20°S (left column), 20°S-20°N (center column) and 20°N-90°N (right column), for different pressure levels as labeled for each row, from 1.0 to 1000 hPa.







**Figure 8.** Normalised differences in root mean square error (drmse) in the forecast temperature field (K) with the new and default ozone schemes, as a function of leadtime. The forecast model runs cover 1-year period, with 10-day forecasts from August 2012 to July 2013. The figure shows three latitudinal bands, 90°S-20°S (left column), 20°S-20°N (center column) and 20°N-90°N (right column), for different pressure levels as labeled for each row, from 1.0 to 1000 hPa.





**Figure 9.** Mean error in the temperature field (K) compared to the operational ECMWF analyses, as a function of leadtime for the forecast experiment using the new BMS prognostic ozone (black) and the experiment using the MACC ozone climatology (red) in the radiation scheme. The two 10-day forecast model runs cover 1-year period, from August 2012 to July 2013. The figure shows three latitudinal bands, 90°S-20°S (left column), 20°S-20°N (center column) and 20°N-90°N (right column), for different pressure levels as labeled for each row, from 1.0 to 1000 hPa.

**Figure 10.** Normalised differences in root mean square error (drmse), as a function of leadtime, in the forecast temperature field (K) between the experiment using the new interactive ozone scheme and the experiment using the default ozone climatology within the radiation code. The forecast model runs cover 1-year period, with 10-day forecasts from August 2012 to July 2013. The figure shows three latitudinal bands, 90°S-20°S (left column), 20°S-20°N (center column) and 20°N-90°N (right column), for different pressure levels as labeled for each row, from 1.0 to 1000 hPa.

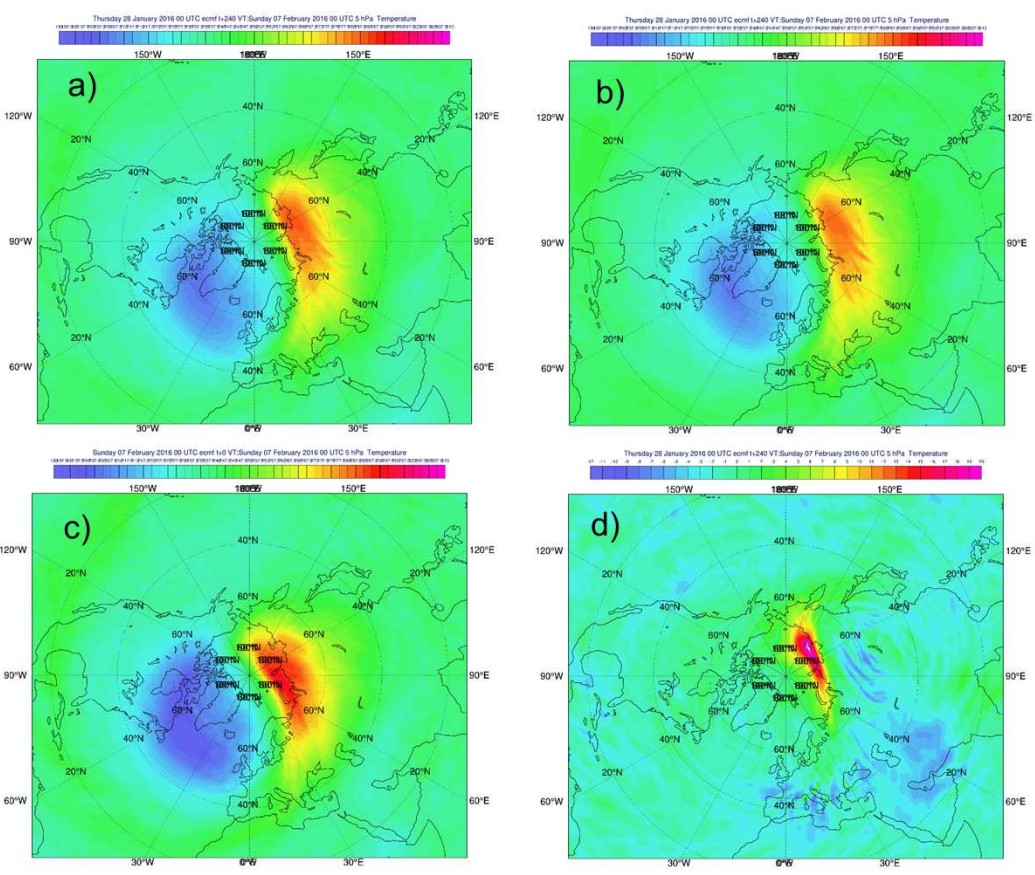

**Figure 11.** Temperature field on the $7^{th}$ of February 2016 at 5 hPa from a) the forecast experiment using the new BMS ozone scheme, b) the forecast experiment using the default ozone scheme, c) the operational analysis. The differences between both forecast experiments are shown in panel d). For the forecast experiments the field shown corresponds to day 10 of the forecast initialised on the $28^{th}$ of January 2016.



**Figure 12.** Cross sections for the change of RMS error in the horizontal v wind velocity for different lead times (24 to 240 hours) for the period $15^{th}$ December 2015-$15^{th}$ February 2016. Each experiment, one using the new ozone scheme interactive with radiation and one using the default scheme, is compared to the ECMWF operational analysis as reference for normalising the error difference; this figure displays the differences between both experiments compared to the reference. Hatched regions indicate where results are statistically significant to the 95 % confidence interval.

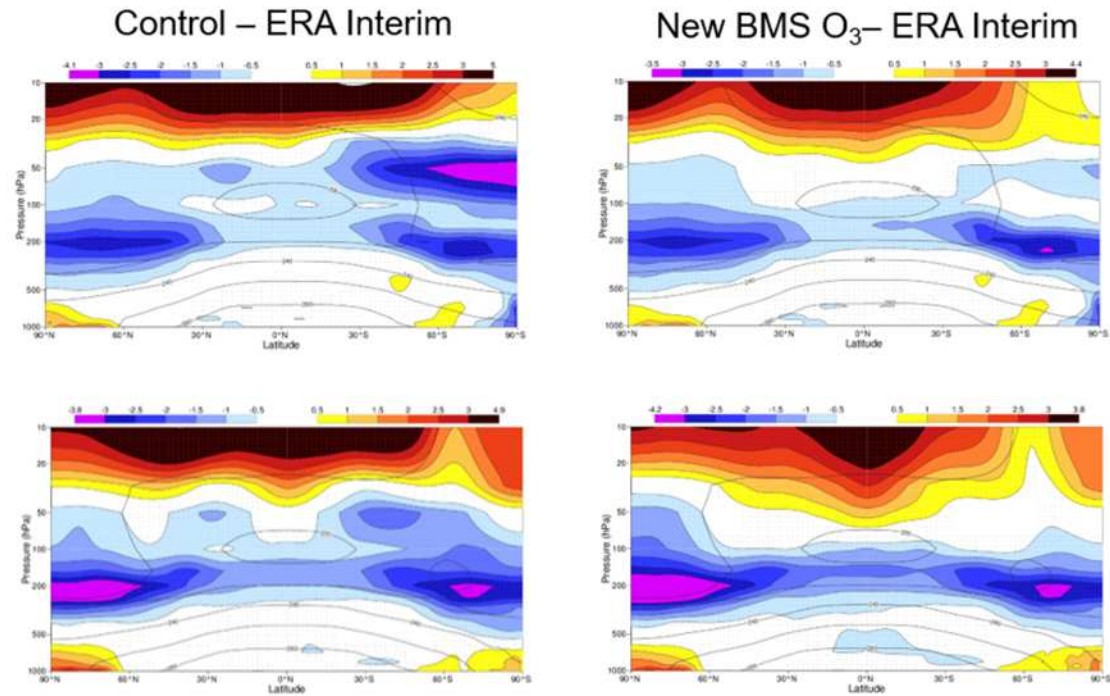

**Figure 13.** Zonal average of temperature differences compared to ERA-Interim for two long range model experiments, one with the default model configuration in which the radiation scheme sees the ozone climatology (left panels), and another one in which the new ozone scheme is used and made interactive with radiation (right panels). Upper panels show DJF differences, and lower panels MAM differences. The period covered is 1981-2010 and the model version used is Cy41r1 with an horizontal resolution T255 and 91 vertical levels up to 0.01 hPa. Dotted areas indicate regions in which the differences are statistically significant at 95% confidence level.



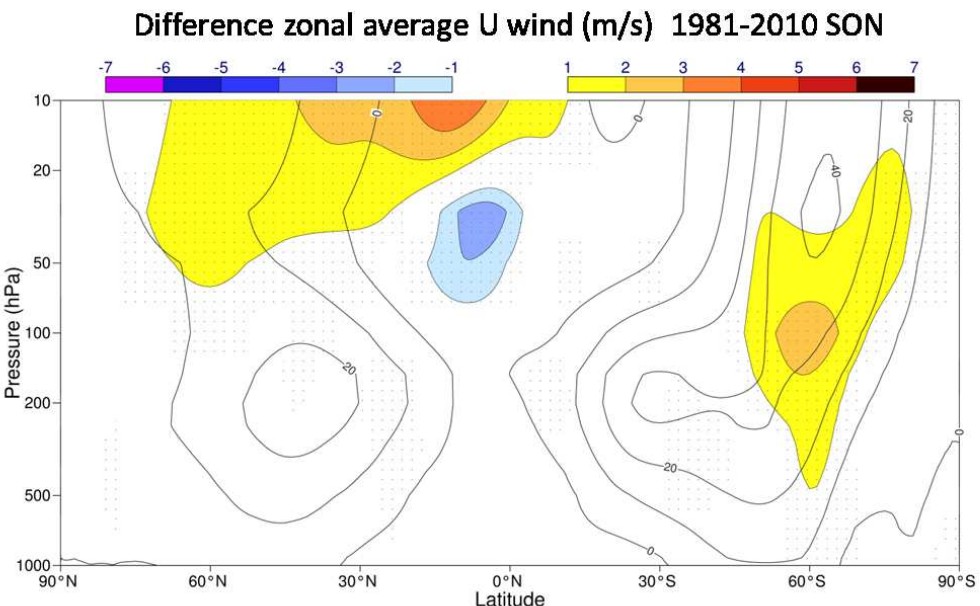

**Figure 14.** Zonal average of differences in the zonal wind u (m/s) between the long-range experiment using the new ozone scheme and a long-range experiment using the default configuration, for the September-October-November (SON) season. Experiments and model configurations used here are as in Fig. 13. Hatched areas indicate regions where the differences are statistically significant at the 95 % confidence interval.





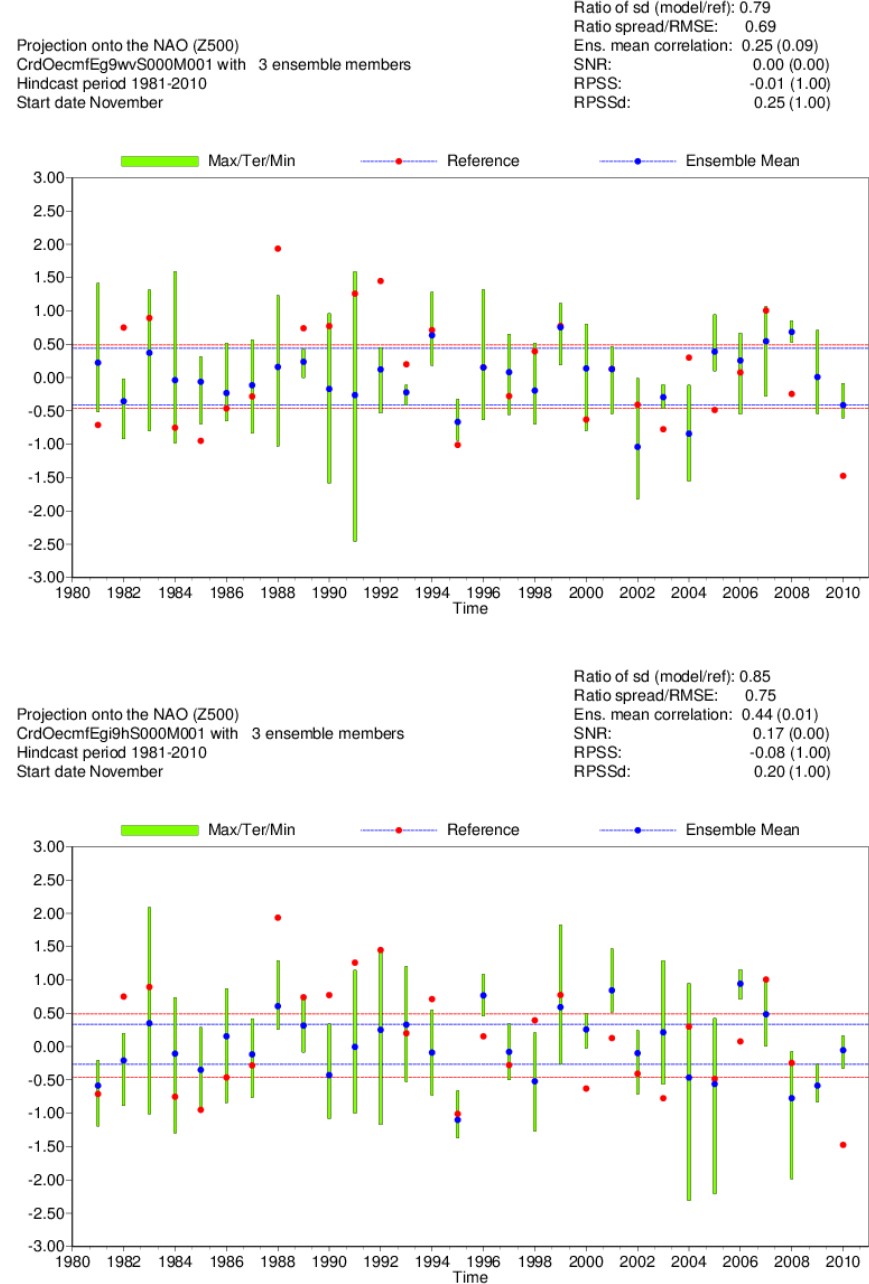

**Figure 15.** Time series of the winter (DJF) NAO index for the long-range experiment using the default configuration (top) and the one using the new BMS ozone (bottom). This plot shows the correlation between the model experiments and ERA-Interim. Red dots represent the NAO index value from ERA-Interim reanalysis, while blue dots correspond to the ensemble mean of the corresponding model run; the green bars show the ensemble variability for each year NAO index. The NAO index is calculated as the first empirical orthogonal function (EOF) of the geopotential height Z500 over the North Atlantic sector.

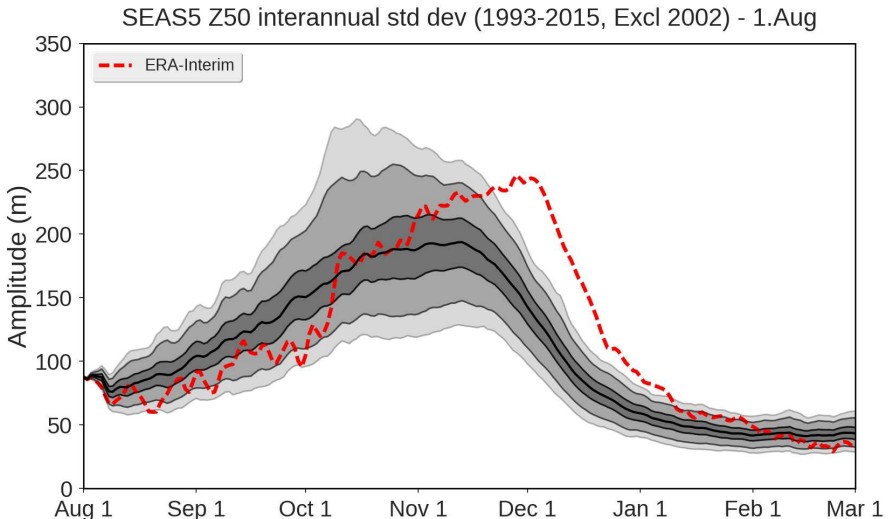

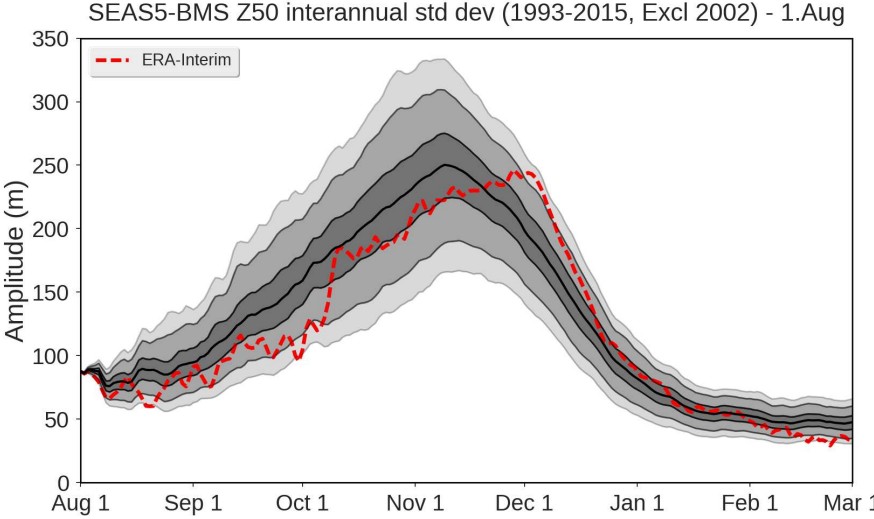

**Figure 16.** Polar cap geopotential height interannual standard deviation for the SEAS5 default simulation (top) and the SEAS5 simulation using the BMS ozone model (bottom); simulations cover the period 1993-2015, $1^{st}$ of August start dates have been used. The year 2002 has been excluded. The red dotted line shows corresponding values from ERA-Interim.

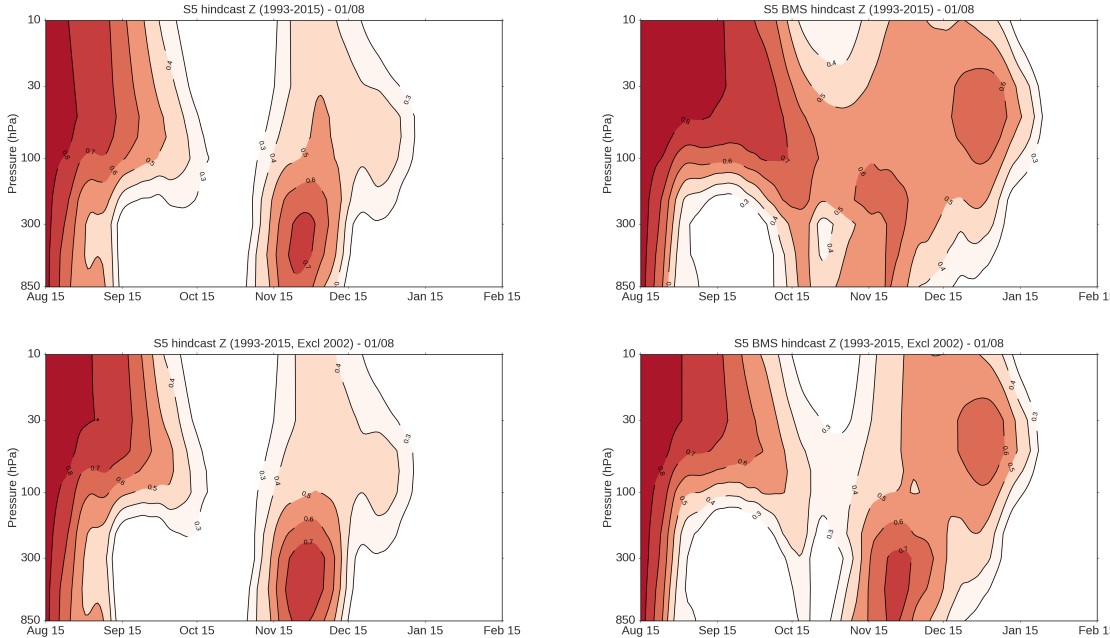

**Figure 17.** Correlation values between 30-day mean ensemble mean SH polar-cap averaged geopotential height for the seasonal experiments and 30-day mean ERA-Interim SH polar-cap averaged geopotential height, as a function of pressure level and calendar day. Seasonal experiments are from SEAS5 default configuration (left column) and from SEAS5 with BMS ozone (right column). The period shown is 1993-2015 (top row) and 1993-2015 excluding year 2002 (bottom row). Coloured areas indicate statistically significant correlations at the 95 % confidence level. Darkest reds indicate higher correlations, countour labels increase by 0.1.

**Table 1.** Medium-range (10-day forecasts) IFS model experiments using the default IFS Cariolle v2.9 ozone scheme ('CD'), or the new BMS ozone scheme ('BMS'). For each experiment information on model version, resolution and period covered is included, as well as whether prognostic ozone was interactive with radiation is also included.

| Exp | IFS | Resol | O$_3$ model | period | interactive |
|---|---|---|---|---|---|
| exp001bms | CY41R1 | T511 L91 | BMS | 1.Aug.2012- 1.Aug.2013 | no |
| exp001cd | CY41R1 | T511 L91 | CD | 1.Aug.2012- 1.Aug.2013 | no |
| exp002bms | CY41R1 | T511 L91 | BMS | 1.Aug.2012- 1.Aug.2013 | yes |
| exp002cd | CY41R1 | T511 L91 | CD | 1.Aug.2012- 1.Aug.2013 | yes |
| exp2016bms | CY41R1 | T511 L91 | BMS | 15.Dec.2015- 15.Feb.2016 | yes |
| exp2016cd | CY41R1 | T511 L91 | CD | 15.Dec.2015- 15.Feb.2016 | no |
| exp2002bms | CY38r2 | T159 L91 | BMS | 1.Aug.2002 -31.Dec 2002 | no |
| exp2002cd | CY38r2 | T159 L91 | CD | 1.Aug.2002 -31.Dec 2002 | no |





**Table 2.** Seasonal experiments using the new ozone scheme ('BMS'), or the default ozone configuration in which the radiation sees an ozone climatology ('CLIM').
For each experiment information on model version, resolution, period covered, as well as the number of ensemble members is also included.

| Exp | IFS | Resol | O$_3$ model | period | members |
|---|---|---|---|---|---|
| exp10bms | CY41R1 | T255 L137 | BMS | 2001- 2010 | 5 |
| exp10clim | CY41R1 | T255 L137 | CLIM | 2001- 2010 | 5 |
| exp30clim | CY41R1 | T255 L91 | CLIM | 1981- 2010 | 3 |
| exp30bms | CY41R1 | T255 L91 | BMS | 1981- 2010 | 3 |
| SEAS5ctr | CY43R1 | TCo319 L91 | CLIM | 1993- 2015 | 25 |
| SEAS5BMS | CY43R1 | TCo319 L91 | BMS | 1993- 2015 | 25 |