# Peer review of "A stratospheric prognostic ozone for seamless Earth System Models: performance, impacts and future"

_Atmospheric Chemistry and Physics, 2020_

## Referee Comment (RC1)

**Review of the manuscript "*A stratospheric prognostic ozone for seamless Earth System Models: performance, impacts and future*" by B.M. Monge-Sanz et al. 2021**

This study discusses an implementation of a previously described and tested computationally efficient ozone chemistry scheme in the ECMWF operational NWP system. This new scheme replaces the default Cariolle and Déqué formulation. The new BMS scheme models ozone chemical tendencies as sum of linear terms dependent on local ozone concentration, temperature and overhead ozone column anomalies relative to a climatology, with tendency coefficients derived from a full chemistry model. One of the major advantages of the BMS scheme over CD is that heterogeneous chemistry is implicit in the tendency coefficients. The paper demonstrates that the ECMWF system equipped with BMS produces ozone fields that are in better agreement with independent data, including in anomalous polar winter/spring seasons such as during the 2002 Antarctic ozone hole split. Furthermore, when ozone fields are fed back into the radiation code, there are substantial improvements in the representation of stratospheric and even tropospheric dynamics leading to improved forecast skills.

The paper is clearly written, and the results are convincing for the most part. Quantification of impacts of improved representations of stratospheric ozone on forecast meteorology has been a long-standing open problem in our field. I think this work will be of interest to the NWP and S2S communities. I recommend this paper for publication subject to some revisions. I don't have any major criticisms of the analysis, but I do have a number of specific comments. In particular, many of the figures are barely legible, and many of them are not properly captioned. Figure 15 is the most striking example; it contains a large number of acronyms and numbers, almost none of which are explained or discussed anywhere. I also feel that the selection of metrics used (e.g. in Figure 12) needs some stronger justification.

**Specific comments**

L14. What is this correlation with? ERA-Interim?

L20 "in timescales" → "on timescales"

L45. Why "not yet available"? We have full chemistry models including CCMs. The tools are out there. I think one issue is computational feasibility of having a chemistry model drive ozone in NWP applications and atmospheric reanalyses, although in the latter case there are a few reanalyses that use full chemistry (albeit without radiative feedback, e.g. CAMS, BRAM2)

L73. "previous linear ozone model," → model**s**

L75 expand SLIMCAT (first use)

L85. Is "new BMS" different than BMS discussed above? Why "new"?

L100. "evolves in time according to the following equation". That's a little imprecise. This equation contains chemistry only; the full evolution equation will also have transport terms. I suggest rephrasing.

L106. So they implicitly assume a certain level of ODS concentrations, right? Please explain.

L125-126. I don't think I understand how this type of processes is not missed in Eq. 1. Which of the four terms includes it? The coefficients there are latitude dependent climatological so they don't know about a polar filament venturing out of the vortex.

Section 2.2. How are the experiments initialized? From assimilated ozone? If so what data are assimilated?

L145. Table 1 only lists experiments at T511 and T159 and only at 91 levels. Other resolutions are for seasonal forecasts listed in Table 2 and discussed in the next subsection.

L168-176. This paragraph lacks any references to published core papers on those data sources, data versions, or on uncertainty estimates.

Sections 3.1 and 3.2 (Figures 1 & 2). Consider also showing and discussing difference panels

L222. "Latitudes over 62°S". I think "latitudes south of 62°S" would be clearer if that's what is meant here.

LL224-225. I'm not sure if this statement is supported by Flemming et al. Fig.16, the relevant panel of which I pasted below. Looking at the Flemming et al. figure, in the period 2003-2010 CAMSiRA is biased high by up to 20 DU (not 30) during the summer months, but the bias is quickly reduced during September (onset of the ozone hole) and, actually, becomes negative in October-December. On the other hand, Fig. 3 of this manuscript indicates that in the later months about half of the BMS runs are higher and half are similar to CAMSiRA. I think that the argument would be more convincing if BMS and CD total ozone were compared directly to observations in addition to showing the plots in Fig. 3. Also, what is the average difference between BMS and CAMSiRA in the summer months? Is it comparable to what the Flemming et al. plot shows?

[Figure]

LL235-238. Actually, because all the BMS runs that do close the ozone hole do so within a week or so around 30 November I don't know how much can be said about the variability of ozone hole closure dates.

Fig.5 caption.  What are the shaded areas?

LL297-301. I would like to see a more quantitative analysis here. What is the sensitivity of polar winter/spring stratospheric temperature to differences in ozone of the order discussed here (such as between climatological or CD schemes and BMS)? Why should we believe that the cold bias exhibited by the "past versions of the operational ECMWF" model is related to ozone rather that other features of those versions of the model, e.g. gravity wave drag, resolution, representation of subsidence within the polar vortex, etc.? These couple of sentences read as very speculative. I think that either a more substantive analysis is needed (similar to Fig. 7), or these sentences should be deleted.

LL311-321. Some of these improvements are very impressive. Nice!

Fig. 11 the labels on the maps and color bars are unreadable. The font has to be made much larger. It also looks like in several places multiple labels are printed on top of one another.

[Figure]

Fig 11 caption. Please state what is getting subtracted from what in panel (d). Also, I think the difference plot would be punchier with a red-blue color scheme.

Fig. 12. Please increase the resolution. The figure is blurry. Also, please add units to the color bar and to the caption.

LL364-371. It's very nice to see this downward signal propagation with increasing lead time but I would like to see some expanded discussion here: I see RMS error improvements 0.01–0.05 (m/s?). How does that compare to typical absolute meridional wind velocities in the regions where these improvements occur? Why were meridional winds chosen as a metric here? Finally, I would like to see a similar discussion of geopotential heights, as this is a more commonly used metric. GPH is shown for the seasonal experiments so I hope it shouldn't be too hard to discuss them here as well.

Fig. 13. Again, it needs a much higher resolution and much larger font. The caption mentions dotted regions indicating significance but there doesn't appear to be any (is that because of poor resolution?).

L376. Are the initial conditions for these experiments from ERA-Interim?

L383. Why SON, not DJF and/or MAM as in the temperature discussion? Is there a rationale behind this particular selection of seasons for different metrics?

L386. Does that constitute an improvement? Would it be possible to verify it against the reanalysis as it was done for temperature?

Fig. 14. The hatching is barely visible. I had to magnify the figure to see it clearly. Also, please consider refining the difference intervals. As it is now, there are large white (i.e. consistent with zero difference) patches marked as statistically significant, which looks strange. Are some of the very small (within +/- 1m/s) differences really significant at 95%?

Fig. 15. I find this figure very hard to read. The caption states "This plot **shows the correlation** between the model experiments and ERA-Interim. Red dots represent the NAO index value from ERA-Interim reanalysis, while blue dots correspond to the ensemble mean of the corresponding model run". This is confusing: either the plot shows the correlation of the NAO indexes, or it shows the indexes themselves. The entire figure looks like it was generated by some standard operational diagnostic software. The ECMWF inner circle may be familiar with the format, but the general reader is not. What are *SNR*, *RPSS*, and *RPSSd*? What does "*CrdOecmfEgi9hS000M001 with 3 ensemble members*" mean? What is "Ratio spread"? What does "Max/Ter/Min" mean? The legends show blue and red dots on top of blue dashed lines, a combination that doesn't correspond to anything in the plots, etc., etc. None of this is explained, or discussed in Section 4.3.1. I really can't tell if this figure shows anything relevant. From the text I learned that the correlation goes from 0.25 up to 0.44 and that's all I got from these plots and discussion. Please replace it with a simple meaningful and properly captioned figure that supports the claims made in this section.

LL395-398. This explanation, while plausible, is not demonstrated. Please, either show that this is indeed the mechanism (preferred) or drop these sentences. It would be nice to see this part expanded and the results demonstrated in detail.

Fig. 16. There are three shaded variability envelopes in these plots and no explanation what variability measure and thresholds are used. Also, please explicitly state that the ERA-Interim line represents the average over the same years if that's the case.

Fig. 17. Please, increase the font size, especially in the contour labels

The experiment names from Tables 1 and 2 are not used very much in the text. Instead, the text uses descriptive names, which is helpful, but it would be good if both were stated. For example, the Figure 1 caption could say "simulated with the new ozone scheme in IFS ("**exp001bms**"; middle) "

---

## Referee Comment (RC2)

**Review** of "A stratospheric prognostic ozone for seamless Earth System Models: performance, impacts and future" by Monge-Sanz et al.

In this study the authors evaluate the implementation of a new stratospheric ozone model in the ECMWF system, both in terms of the fidelity of the simulated ozone and in terms of the resulting ozone feedbacks on radiation and dynamics in the model. Using a combination of both medium-range (10 day-long) and long-range (7 month) ensemble forecasts, the authors show that the new ozone linearization produces enhanced results, compared to the default ozone implementation in the IFS system. Quite importantly, this improved performance is exhibited in extreme dynamical conditions such as stratospheric sudden warmings, with improved coupling to the surface (manifest in correlations in the NAO). Overall, the manuscript is well motivated, well written and thorough. Therefore, it is, in my opinion, deserving of publication pending minor revisions. At the same time, however, there are several points where certain details of the new ozone implementation need to be clarified, specifically as they compare to the "default" implementation. These clarifications are needed as they may impact the extent to which certain purported "improvements" in the ozone tracer (often described throughout the manuscript as resulting from a more complete implicit incorporation of heterogenous chemical processes) may actually reflect other differences between the schemes (i.e. resolution of the corresponding photochemical box model, etc.).

**Major Comments:**

**1: I find it somewhat misleading when the authors claim that the BMS scheme is a fundamentally "new model" compared to the CD scheme. Perhaps this is just an issue of semantics, but if my understanding of the differences is correct isn't the main point the fact that the tendencies for the BMS scheme are derived from a box model with updated chemistry (i.e. one that includes heterogenous chemical effects), unlike the default scheme where these processes need to be included as an additional (crudely parameterized) term? It seems to me, therefore, that it's not really true that the linearization itself is fundamentally changed (except for the exclusion of a term that is now not needed); rather, what is new is the source of the tendencies that are being input to the linearization. I suppose that this modified input (now from a 3-D CTM with an improved chemical mechanism) constitutes part of the model and therefore represents a "new model" but I would think a more conventional interpretation of "new model" is one that involves additional terms to equations (1-2) and/or changes in functional dependencies (i.e. not simply changes in the input). My suggestion here is to be a bit more clear that what is meant by a "new model" is really mainly a reflection of new tendencies that derive from a parent GCM that now happens to incorporate heterogenous chemical processes. The functional form of the linearized ozone tracer itself has not really changed.**

**2. Since one of the main points of the study is to demonstrate the improvement of the new ozone tracer compared to the default approach, more details are needed about the exact differences between the BMS and CD approaches. Therefore, while it may be true that some of the details that are wanting can be found in Monge-Sanz (2011) I feel that a subset can be repeated here. The reason for this is that the authors tend to attribute most of the improved**

ozone performance to the incorporation of implicit heterogenous chemical effects but there may be other factors at play as well.  In particular, what is the vertical resolution of the 2D photochemical model in the default ozone implementation?  The coefficients for the BMS scheme are derived from the box model version of a 3D CTM.  They are calculated at 24 levels (and then presumably interpolated somehow to the native vertical grid of the IFS forecasts, although this detail is not provided).  Are the coefficients from the default ozone linearization also available at these levels?  If the treatment of vertical gradients is still more simple in that model it is not obvious to me that the BMS and CD main differences reside in the treatment of heterogeneous chemistry.  Please, I strongly suggest the authors add more details to Section 2.1 that address resolution (and forcing set) differences between the box models separately used to produces the coefficients for these different ozone tracers.  Moreover, speculate on how these aspects of the ozone tracer linearization may be contributing to the differences seen (in the conclusions).

Another point that needs clarification in Section 2.1 is the meteorological fields that are used to initially derive the coefficients in both the BMS and default ozone linearization.   What is the limitation of using coefficients based on box model runs using input meteorology for one representative year?  If the results are based on coefficients on a year with mild temperatures one might expect that this would restricts the ability of the scheme to reproduce realistic ozone loss during colder winters and vice versa.  Is this an issue for both ozone tracers?

**3. Major Comment: One issue that is not discussed is the increased computational expense of the BMS scheme, which is introduced by needing to run the full 3-D CTM (and feed into the box model of the CTM to produce the coefficients).  While this may be only a "one time" step it nonetheless is more expensive than simply running a 2-D box model using, for example, observed ozone climatologies that already exist.  Indeed, the authors even suggest that it may be needed to constantly run the full CTM in order to provide a new "carefully chosen climatology term…from an updated run of the parent full-chemistry CTM."  While that certainly will be good (to the extent that the CTM itself does a good job at reproducing observed ozone) it is nonetheless not a small computational ask.  I suggest that the authors discuss this trade-off between performance and computational cost in the conclusions.**

Minor Comments:

Line 29: A semi-colon (not comma) is needed after "exothermic reactions".
Line 99: A "the" is needed in "can simulate time".
Line 103: More description here is needed regarding the "tendencies derived from the full-chemistry CTM runs".  My specific concern regards ambiguities about the forcings and boundary conditions used for the CTM runs from which these full tendencies are derived.  In particular, over what time period are these CTM runs performed?   More generally, what are the boundary conditions and compositional forcings used in the CTM simulations and how consistent are these with those used in the IFS forecast ensembles?
Line 122: The "to" after "sunlight" should be removed.

Line 133: I'm not sure I agree with this statement beginning with "In addition, the coefficients…". While it may be true that the coefficients are derived from a 3D model, what is used in the BMS scheme are the zonally averaged fields. mPlease reconcile this discrepancy.

Line 141: I am not sure I agree with the sentence "The IFS configuration in each pair … differs only in the scheme used to model stratospheric ozone." Is this really true? Later on in the manuscript, you write that the old scheme does not include radiative feedbacks from ozone so isn't this a key difference with the new BMS scheme? I understand that you ran experiments with the BMS scheme in which feedbacks are/are not incorporated but please indicate here that this is another key difference between the schemes.

Line 145: Given that the box model coefficients are inferred from a much coarser CTM output (7.5 horizontal resolution and 24 levels, according to Monge-Sanz et al. (2011)) how do you account for the increased resolution of the native model grid? Do you simply interpolate? Wouldn't it be worth running the CTM at a higher resolution (albeit with the additional expense) to get more realistic coefficients?

Line 164: Remove "an" before "horizontal resolution".

Line 166: Are these ensembles also launched in May and November of each year over the period 1981-2010? It is not clear when these are initialized.

Line 171: An "and" is needed before "the National Oceanic …".

Line 185: Experiment names (from Table 1) need to be indicated as it is not clear if the only difference here is the linearization scheme or the interactions with the radiation. Simply providing the experiment names will help clarify this. In particular, are you comparing exp001bms and exp001cd or exp002bms and exp001cd?

Line 200: Typo in "characterstics".

Line 209: What do you mean exactly by "the more realistic link with temperature in the new scheme"? Is this because of the implicit incorporation of heterogenous chemistry in the coefficients or are you comparing the schemes with and without feedbacks to radiation? Again, clarification about which exact experiments are being analyzed is needed. See my previous comment.

Line 216: Is there a reason why these experiments, which are described earlier, are being described again? Why the need to repeat these experiment details? Or are these different runs?

Line 229: I'm not sure "manifests" is the right word here.

Line 261: I'm not convinced that the use of the "a recent observation-based climatology" would necessarily be ideal. Isn't one of the main improvements afforded from the BMS scheme the fact that it better ensures internal consistency between the gas phase and heterogenous chemistry? To this end, therefore, wouldn't replacing the climatology term with one derived from observations break consistency with the other terms (which are derived from the CTM)? If consistency is a desirable feature, I would think that using a combination of model and observationally derived coefficients would be problematic.

Line 285: Again, why can't this comparison be cleaner? You should do either BMS w/wout radiative feedback (or using CD). What does that show? I suggest the authors replace this text here with a discussion of those results since this is not a proper apples-to-apples comparison.

Line 480: This is a more of comment that could be addressed in the conclusions but have you considered modifying BMS coefficient calculation to account for zonal variations in ozone? If this

is what is already meant by "the BMS scheme coefficients could be provided in a 3D grid" then please clarify that, indeed, this is the case.

---

## Author Comment (AC1)

**Authors' responses for manuscript acp-2020-1261 "*A stratospheric prognostic ozone for seamless Earth System Models: performance, impacts and future*" by Monge-Sanz et al. https://doi.org/10.5194/acp-2020-1261**

We sincerely thank both Reviewers for their positive, thorough and constructive reviews, and for the appreciation of the usefulness of this study and its results.

We have addressed all their comments and edited the manuscript accordingly. Please find here below our detailed responses. Reviewers' comments are in **bold**, and our responses in normal font; when needed *italics* are used to indicate text edited/added for the revised version. Also please note that, unless otherwise stated, the numbering for lines and figures in our responses refers to the original numbering (i.e. numbering in the Reviewers' comments, in case of ambiguity both new and old numbering are mentioned).

We would also like to thank the handling Editor for useful initial comments on the manuscript, which we also incorporated at an earlier stage.

**Reviewer 1:**

**Comment: The paper is clearly written, and the results are convincing for the most part. Quantification of impacts of improved representations of stratospheric ozone on forecast meteorology has been a long-standing open problem in our field. I think this work will be of interest to the NWP and S2S communities. I recommend this paper for publication subject to some revisions. I don't have any major criticisms of the analysis, but I do have a number of specific comments. In particular, many of the figures are barely legible, and many of them are not properly captioned. Figure 15 is the most striking example; it contains a large number of acronyms and numbers, almost none of which are explained or discussed anywhere. I also feel that the selection of metrics used (e.g. in Figure 12) needs some stronger justification.**

We thank this Reviewer for their positive evaluation and all their constructive comments, which have contributed to enhance the manuscript. The quality of the figures is improved and captions clarified as necessary for the final manuscript version. We address specific corresponding comments below. In particular for Figure 15 and Figure 12, see our responses and related changes in the specific comments below in this document.

**Specific comments**

**L14. What is this correlation with? ERA-Interim?**

Yes, ERA-Interim. This information was in the main text and corresponding figures, we have now also added it to the abstract. Thanks.

**L20 "in timescales" → "on timescales"**

Done, thanks.

**L45. Why "not yet available"? We have full chemistry models including CCMs. The tools are out there. I think one issue is computational feasibility of having a chemistry model drive ozone in NWP applications and atmospheric reanalyses, although in the latter case there are a few reanalyses that use full chemistry (albeit without radiative feedback, e.g. CAMS, BRAM2)**

Here we mean tools that can be used across all timescales, from weather to climate. These tools do not exist yet, as CCMs cannot be run at the resolutions and processing times required by weather models (neither at medium-range nor at seasonal timescales). And it is this later type of models the one required to answer questions related to links with weather extremes in a more accurate manner. The main reason to develop simplified chemistry models is precisely to have tools that can provide realistic chemistry fields for models and applications that otherwise would not include the same information. Also, one specific point we demonstrate in this paper is the need for the radiative feedbacks with ozone to improve model predictions.

This was already developed in the subsequent two paragraphs in the main text. We have now also edited this sentence to clarify its meaning and avoid any misunderstanding. Edited sentence: *"Stratospheric ozone research has been very active during the past 30 years (WMO, 2019). Nowadays, the most pressing questions in this field concern the links between the evolution of the ozone layer, climate change and weather extremes. To accurately tackle them, we need tools that can seamlessly operate across timescales from weather to climate."*

**L73. "previous linear ozone model," → models**

We prefer to keep the current phrasing as it reflects better what we mean ("..*any* other previous linear ozone model").

**L75 expand SLIMCAT (first use)**

Many past publications related to this model also had to clarify this during review: SLIMCAT is not an acronym, it is a model name with no spelling out correspondence.

**L85. Is "new BMS" different than BMS discussed above? Why "new"?**

It is the BMS scheme discussed above. Here we meant that it is a new scheme compared to existing ones, and that it is also a new scheme implemented in the ECMWF model.

The sentence has been edited to make it clearer.

**L100. "evolves in time according to the following equation". That's a little imprecise. This equation contains chemistry only; the full evolution equation will also have transport terms. I suggest rephrasing.**

Thanks. We have rephrased this sentence to avoid misunderstanding. It now reads: *"…can simulate the time evolution of ozone by including an advected tracer for which the local concentration f (i.e. the local net ozone chemical production minus loss) evolves in time according to the following equation:…"*

**L106. So they implicitly assume a certain level of ODS concentrations, right? Please explain.**

The full-chemistry CTM used to compute the BMS scheme includes ozone depleting species and corresponding reactions. Therefore, there is no need to assume any particular ODS levels once the BMS scheme is computed and implemented in the GCM. This is one of the main novel advantages compared to the CD scheme.

**L125-126. I don't think I understand how this type of processes is not missed in Eq. 1. Which of the four terms includes it? The coefficients there are latitude dependent climatological so they don't know about a polar filament venturing out of the vortex.**

The additional heterogeneous term used by the CD scheme in the ECMWF system, artificially forces ozone depletion to occur only where temperatures are below the threshold it fixes (i.e inside the vortex). As argued in the main text, this is not a realistic scenario for important atmospheric situations, including cases where, e.g. due to vortex filamentation, air processed inside the vortex has reached sunlit areas outside the vortex.

The BMS model can better simulate these effects because for temperature values, and partial ozone column values, for which heterogenous depletion occurs in the full-chemistry model, the coefficients in the BMS do not restrict the location where depletion happens. Therefore, the BMS model allows heterogeneous chemistry processes to occur outside the vortex.

In the BMS model, all coefficients have been obtained taking into account all heterogenous chemistry included by the CTM, that is why we specify that heterogeneous chemistry is embedded in all the terms in Eq.1. This is why the BMS model does not need any specific additional term to be able to simulate ozone loss due to heterogeneous chemistry.

We have added a sentence to help clarify this aspect in the text, Section 2.1 (second paragraph from end of Section): "*Since the BMS scheme coefficients have been obtained taking into account all heterogenous chemistry included by the CTM, heterogeneous chemistry is embedded in all the terms in Eq.1 and not restricted to limited regions and times as in the CD scheme.*"

**Section 2.2. How are the experiments initialized? From assimilated ozone? If so what data are assimilated?**

Medium-range experiments are initialised from operational ECMWF analyses (line 146 in original numbering), while ERA-Interim is used to initialise the long-range experiments (line 160 in original numbering). Our simulations include initial spin-up periods so that ozone values for the periods we analyse depend on the ozone model we use and not on the ozone initialisation values. Same initialisation and spin-up is applied to experiments using both the BMS and the CD scheme.

We have now added information about the spin-up to Section 2.2.

**L145. Table 1 only lists experiments at T511 and T159 and only at 91 levels. Other resolutions are for seasonal forecasts listed in Table 2 and discussed in the next subsection.**

Thanks, this has now been edited to make it consistent with paragraphs and tables in the current version of the manuscript.

**L168-176. This paragraph lacks any references to published core papers on those data sources, data versions, or on uncertainty estimates.**

Many thanks for this comment, references have now been added to this paragraph.

**Sections 3.1 and 3.2 (Figures 1 & 2). Consider also showing and discussing difference panels**

We have now added differences for the 1-year runs we show in Fig. 1, this is now included as Fig.2 (new numbering). We have also added the corresponding discussion into the main text to include the additional information on these differences which nicely confirm our results and discussion of Fig. 1.

Added text (after line 190, old original line numbering): *"The distribution of differences between these model runs and the sonde observations (model-observations) are shown in Fig. 2. The largest differences for the BMS model run (up to 3 mPa) are found in August between 100-200 hPa and in Sep-Oct between 20-50 hPa; for all other months and regions differences are small (within +- 1.5 mPa). However, for the CD model run positive biases (of up to 4 mPa) between 100-200 hPa persist over the first 6 months, and differences with observations are larger between 30-100 hPa from Sep-Apr (negatively biased, with values of up to -5 mPa). […] In the CD model run differences centred around 20 hPa are also larger than in the BMS run from Mar-Jul. "*

Differences for Figure 2 (old numbering, Fig.3 now) are similar in structure and magnitude to differences corresponding to Fig.1, and also confirming the structure of differences that can be seen by inspection of the monthly ozone profiles in Fig. 2 (old numbering, Fig.3 now). We add one sentence to make this clear: *"Differences against observations for these model runs exhibit similar structure and values to those in Fig2."*

**L222. "Latitudes over 62oS". I think "latitudes south of 62oS" would be clearer if that's what is meant here.**

Yes, that is what is meant, we have edited it to clarify, thanks.

**LL224-225. I'm not sure if this statement is supported by Flemming et al. Fig.16, the relevant panel of which I pasted below. Looking at the Flemming et al. figure, in the period 2003-2010 CAMSiRA is biased high by up to 20 DU (not 30) during the summer months, but the bias is quickly reduced during September (onset of the ozone hole) and, actually, becomes negative in October-December. On the other hand, Fig. 3 of this manuscript indicates that in the later months about half of the BMS runs are higher and half are similar to CAMSiRA. I think that the argument would be more convincing if BMS and CD total ozone were compared directly to observations in addition to showing the plots in Fig. 3. Also, what is the average difference between BMS and CAMSiRA in the summer months? Is it comparable to what the Flemming et al. plot shows?**

[Figure]

[Figure]

Thanks, you are right that for the period considered in the comparison in this part of our paper (2003-2010), CAMSiRA is up to 20 DU too large, not 30 DU. We have now corrected this in the manuscript.

In agreement with the point you raise, we have also rewritten this sentence and added some text to better reflect the way our comparison links to the results shown in the figure of the Flemming et al. paper. The new text reads: "*Total ozone column values from CAMSiRA are known to be too large (by up to 20 DU) over the Antarctic region when compared to independent observations during July-September months for the period considered here (Fig. 16 in Flemming et al., 2017). The new BMS scheme for the July-September months is showing lower minimum values than CAMSiRA, 18 DU lower on average (upper panel in Fig.3), implying more realistic ozone column values. For later months in the year, Fig. 16 in Flemming et al. (2017) shows smaller biases for CAMSiRA, mainly positive during October and mainly negative afterwards. For these later months of the year our results with the BMS scheme show overall agreement with the reanalysis, with minimum Antarctic values within the CAMSiRA range.* "

The CAMSiRA reanalyses are used in our comparison as the observational reference dataset, to allow for full grid data availability on the same grid and times as the model experiments, as we do with other reanalysis data. We additionally already included the indirect comparison with other observation dataset through the discussion of the validation study of CAMSiRA by Flemming et al. to provide the additional information on the known biases in the observational dataset we use.

Yes, the average difference between the BMS runs and the CAMSiRA data over the summer months (Jul-Sep) is 18 DU, comparable with the differences CAMSiRA presents against independent datasets. We are also adding this to the revised text.

**LL235-238. Actually, because all the BMS runs that do close the ozone hole do so within a week or so around 30 November I don't know how much can be said about the variability of ozone hole closure dates.**

The model experiments shown in this figure, both with the CD scheme and the BMS scheme, end on the 30[th] November. We already acknowledged the limitation this means for more accurate discussions on the ozone hole closure (lines 237-238 original numbering). However, from the lower

panel in Fig. 3 (original numbering), we can see that the default CD scheme does not show enough variability in ozone hole closure compared to reanalysis data, while the BMS scheme exhibits variability in better agreement with the reanalysis regarding closure dates. We now have edited this part of the text and added a specific reference to the lower panel in the figure to add clarity.

New edited text: "*The model experiments shown in Fig. 3, both with the CD scheme and the BMS scheme, end on the 30th November, and closure dates beyond this date cannot be analysed. However, from the lower panel in Fig. 3, we can see that the default CD scheme does not show enough variability in ozone hole closure compared to reanalysis data, while the BMS scheme exhibits variability in better agreement with the reanalysis.*"

**Fig.5 caption. What are the shaded areas?**

Shaded areas indicate the density of observation profiles for those altitude ranges for the considered period. This information is now added to the figure caption.

**LL297-301. I would like to see a more quantitative analysis here. What is the sensitivity of polar winter/spring stratospheric temperature to differences in ozone of the order discussed here (such as between climatological or CD schemes and BMS)? Why should we believe that the cold bias exhibited by the "past versions of the operational ECMWF" model is related to ozone rather that other features of those versions of the model, e.g. gravity wave drag, resolution, representation of subsidence within the polar vortex, etc.? These couple of sentences read as very speculative. I think that either a more substantive analysis is needed (similar to Fig. 7), or these sentences should be deleted.**

Sentence in lines 297-298 follows from usual interactions between stratospheric ozone and stratospheric temperature increase (as the one seen during SSWs). Highlighting these mechanisms in our discussion helps to explain why changing the stratospheric ozone description in the model can improve the representation of this type of events.

In lines 299-301, we did not intend to compare against past versions of the operational ECMWF, and we agree that several factors contribute to the improvement with respect to previous model versions. We have edited this sentence to make this clearer. New text: "*Past versions of the operational ECMWF forecast model reproduced SSW events overall weaker (colder) than observed (e.g. Diamantakis, 2014); although not the only factor, this is consistent with the fact that by using an ozone climatology in the radiation scheme, past model versions could not fully reproduce the feedbacks between stratospheric ozone and rapid temperature increases that take place during SSWs.*".

**LL311-321. Some of these improvements are very impressive. Nice!**

Thank you! They are indeed.

**Fig. 11 the labels on the maps and color bars are unreadable. The font has to be made much larger. It also looks like in several places multiple labels are printed on top of one another.**

The readability of these labels will be improved in the revised version.

**Fig 11 caption. Please state what is getting subtracted from what in panel (d). Also, I think the difference plot would be punchier with a red-blue color scheme.**

Panel d) shows the differences between the experiment with the BMS ozone and the experiment with the default ozone (BMS – default). We now add this to the caption.

**Fig. 12. Please increase the resolution. The figure is blurry. Also, please add units to the color bar and to the caption.**

We have increased the quality of the figure. Although the blur effect is not in the original figure file, it might be device dependant when creating the pdf file. Info on units also added to the caption, thanks.

**LL364-371. It's very nice to see this downward signal propagation with increasing lead time but I would like to see some expanded discussion here: I see RMS error improvements 0.01–0.05 (m/s?). How does that compare to typical absolute meridional wind velocities in the regions where these improvements occur? Why were meridional winds chosen as a metric here? Finally, I would like to see a similar discussion of geopotential heights, as this is a more commonly used metric. GPH is shown for the seasonal experiments so I hope it shouldn't be too hard to discuss them here as well.**

Firstly, we need to clarify that the metric we show is the vector wind, not the meridional wind, this was a typo in the text that we have now corrected.

The corresponding plots for the geopotential height metric show a statistically significant improvement for day 10, when using the BMS ozone, at the Arctic vortex edge latitudes (around 60N) from 200 hPa all down to the surface. This is consistent with the improvement we show in the wind field in Fig. 12.

Although geopotential is a good metric for assessing forecast skill at the extra tropics, it is not the best indicator for the upper stratosphere. In addition, being an integrated quantity, it shows greater sensitivity not only to temperature at levels below but also to small changes in total atmospheric mass which occur as the semi-Lagrangian advection in the IFS is not formally conserving. On the other hand, using the vector wind metric for SSW events we can link error reduction to better predictability of the cold air transport that is usually associated with SSW events.

**Fig. 13. Again, it needs a much higher resolution and much larger font. The caption mentions dotted regions indicating significance but there doesn't appear to be any (is that because of poor resolution?).**

Figure quality, including visibility of the dotted areas and font size will be improved in the revised version.

**L376. Are the initial conditions for these experiments from ERA-Interim?**

Yes, as indicated in Section 2.2.2, the long-range experiments in our study are initialised with ERA-Interim fields.

**L383. Why SON, not DJF and/or MAM as in the temperature discussion? Is there a rationale behind this particular selection of seasons for different metrics?**

For wind velocities we focus on the SON season to show the impact on dynamics during the ozone hole season. Differences for other seasons were smaller or not statistically significant. We have edited this sentence to clarify this aspect.

New edited text: *"…shows the zonal averaged differences in zonal wind between the two experiments for the SON season, to assess the impact on wind circulation during the ozone hole season (differences for other seasons were smaller or not statistically significant)."*

**L386. Does that constitute an improvement? Would it be possible to verify it against the reanalysis as it was done for temperature?**

Yes, it constitutes an improvement; over the Antarctic vortex area region we refer to in the manuscript (Fig. 14 in the original numbering), the control run is negatively biased compared to ERA-Interim, and the BMS run is in better agreement with ERA-Interim.

We have edited this paragraph to include this information.

Text added: *"When comparing to ERA-Interim (figure not shown), the control run was negatively biased over this region compared to the reanalysis (up to 2m/s), which appears reduced in the BMS run. The strengthening of winds in the BMS run is also in overall agreement with findings in Son et al. (2008)."*

**Fig. 14. The hatching is barely visible. I had to magnify the figure to see it clearly. Also, please consider refining the difference intervals. As it is now, there are large white (i.e. consistent with zero difference) patches marked as statistically significant, which looks strange. Are some of the very small (within +/- 1m/s) differences really significant at 95%?**

We have replotted this figure with a refined colour scale of +- 4m/s. Now some additional structure is coloured at lower levels, there also remain some statistically significant white areas where values are below +-0.5 m/s. We prefer not to go to lower colour scale values to avoid saturation of the areas where differences are larger.

The quality of the figure, including the visibility of the hatching, is also increased in the revised version.

**Fig. 15. I find this figure very hard to read. The caption states "This plot shows the correlation between the model experiments and ERA-Interim. Red dots represent the NAO index value from ERA-Interim reanalysis, while blue dots correspond to the ensemble mean of the corresponding model run". This is confusing: either the plot shows the correlation of the NAO indexes, or it shows the indexes themselves. The entire figure looks like it was generated by some standard operational diagnostic software. The ECMWF inner circle may be familiar with the format, but the general reader is not. What are SNR, RPSS, and RPSSd? What does "CrdOecmfEgi9hS000M001 with 3 ensemble members" mean? What is "Ratio spread"? What does "Max/Ter/Min" mean? The legends show blue and red dots on top of blue dashed lines, a combination that doesn't correspond to anything in the plots, etc., etc. None of this is explained, or discussed in Section**

**4.3.1. I really can't tell if this figure shows anything relevant. From the text I learned that the correlation goes from 0.25 up to 0.44 and that's all I got from these plots and discussion. Please replace it with a simple meaningful and properly captioned figure that supports the claims made in this section.**

This figure shows the time series of the winter (DJF) NAO index for the long-range experiments, as clearly stated at the start of the caption.

We have edited the figure, also removing unnecessary headers and labels to make it clearer. And we have also included additional information into the caption and corresponding discussion in the main text.

Red dots in this figure represent the value of the NAO index from the ERA-Interim reanalysis (this reference does not change between both panels); blue dots represent the NAO index obtained from the ensemble mean of the corresponding seasonal runs (using the default ozone configuration in the top panel, and the BMS ozone in the bottom panel). The green bars represent the ensemble spread in the NAO index for each year. All this information is already found in the caption. We agree that the second sentence in the caption could have led to confusion and we have now edited it. We have also included additional information as indicated at the end of this reply.

This figure shows one of the main results in terms of the impacts detected with the new ozone. The NAO index is widely used as a proxy for stratosphere-troposphere coupling, the improvement we obtain by changing to the BMS ozone representation shows the impact that a more realistic stratospheric ozone representation has to enhance stratosphere-troposphere coupling in the model, therefore improving possibilities to exploit stratospheric sources of predictability.

The new caption text reads as: "*Time series of the winter (DJF) NAO index for the long-range experiment using (a) the default configuration, and (b) the experiment using the new BMS ozone, together with NAO index values from the ERA-Interim reanalysis. Red dots represent the NAO index value from ERA-Interim reanalysis, blue dots correspond to the ensemble mean of the model runs; the green bars show the ensemble variability for each year NAO index. The NAO index is calculated as the first empirical orthogonal function (EOF) of the geopotential height Z500 over the North Atlantic sector. The correlation value between the ensemble mean NAO index and the reanalysis NAO index time series is also included in each panel, together with the signal to noise ratio (SNR) value for each case. The horizontal lines delimit the main tercile interval for the ensemble mean time series of the model experiments (blue) and ERA-Interim (red).*"

**LL395-398. This explanation, while plausible, is not demonstrated. Please, either show that this is indeed the mechanism (preferred) or drop these sentences. It would be nice to see this part expanded and the results demonstrated in detail.**

In this paragraph we point towards the most plausible physical reasons behind the improvements we see in our model runs for the NAO diagnostic when using the new ozone option. We consider it important to include this type of scientific discussion to explain the changes we obtain in the stratosphere-troposphere coupling behaviour in the model. But it is beyond the scope and resources of this study to demonstrate these mechanisms, which we hope will be part of future work. We have edited the paragraph to make this clearer.

New edited text: *"A more realistic stratospheric ozone distribution (Sect. 3) improves the ozone concentration gradients between the Pole and the equator, modifying the latitudinal heating gradient in the LS region. Plausibly, this affects the altitude distribution of the tropopause in the model, and therefore surface pressure gradients between low and high latitudes and the NAO signal. Future work should be done to fully assess these mechanisms, which is beyond the scope and resources of our current study."*

**Fig. 16. There are three shaded variability envelopes in these plots and no explanation what variability measure and thresholds are used. Also, please explicitly state that the ERA-Interim line represents the average over the same years if that's the case.**

We have 22 years of ERA-Interim data (1993-2015, excl. 2002) from August 1$^{st}$ until March 1$^{st}$. For these, we compute the standard deviation of polar-averaged geopotential height (Z50) for each day of the year (interannual standard deviation); this is the dashed red line.

We perform a similar calculation using SEAS5 data, for which we have 22 years of hindcasts, and 51 ensemble members. For each of the 22 years we randomly select an ensemble member hindcast. We then compute the interannual standard deviation of this randomly selected hindcast time series. We do this 10000 times to produce a probability distribution (shaded envelopes) and we plot the 1%, 5%, 25%, 50%, 75%, 95% and 99% threshold values for each day from August 1$^{st}$ until March 1$^{st}$ (solid lines).

This information is now included in the caption and the main text.

**Fig. 17. Please, increase the font size, especially in the contour labels**

Thanks, this is done for the revised version.

**The experiment names from Tables 1 and 2 are not used very much in the text. Instead, the text uses descriptive names, which is helpful, but it would be good if both were stated. For example, the Figure 1 caption could say "simulated with the new ozone scheme in IFS ("exp001bms"; middle)"**

Thanks, we now mention the name of the experiments when needed along the text to add clarity.

**Reviewer 2:**
**Major Comments:**

**Comment #1: I find it somewhat misleading when the authors claim that the BMS scheme is a fundamentally "new model" compared to the CD scheme. Perhaps this is just an issue of semantics, but if my understanding of the differences is correct isn't the main point the fact that the tendencies for the BMS scheme are derived from a box model with updated chemistry (i.e. one that includes heterogenous chemical effects), unlike the default scheme where these processes need to be included as an additional (crudely parameterized) term? It seems to me, therefore, that it's not really true that the linearization itself is fundamentally changed (except for the exclusion of a term that is now not needed); rather, what is new is the source of the tendencies that are being input to the linearization. I suppose that this modified input (now from a 3-D CTM with an**

improved chemical mechanism) constitutes part of the model and therefore represents a "new model" but I would think a more conventional interpretation of "new model" is one that involves additional terms to equations (1-2) and/or changes in functional dependencies (i.e. not simply changes in the input). My suggestion here is to be a bit more clear that what is meant by a "new model" is really mainly a reflection of new tendencies that derive from a parent GCM that now happens to incorporate heterogenous chemical processes. The functional form of the linearized ozone tracer itself has not really changed.

The differences are not only in the chemistry included in the photochemical model, but more importantly, in the way the heterogeneous chemistry is represented in the parameterisation, i.e. in the formulation of the parameterisation itself.

The number of terms and their nature, i.e. the processes they simulate, is fundamentally different in the BMS model. Not only the number of terms is reduced but they now parameterise all chemistry related to stratospheric ozone. This is a new approach compared to previous linear parameterisations, which omitted the nonlinear ozone heterogeneous chemistry or added an adhoc detached term. It is a significant change compared, not only to the CD scheme version used in the ECMWF model, but compared to any previous linear model for stratospheric ozone.

The BMS model is the first linear model able to eliminate any artificially added term in order to simulate heterogeneous chemistry processes. This is a clear novel approach, not only related to the chemistry in the parent photochemical model but to the goal of having a new more robust way of parameterising stratospheric ozone, making the new model scheme more realistic while keeping a linear approximation.

A few other linear ozone models exist which have also been presented in literature as new models despite maintaining more similarities to the CD scheme approach than the model we present here (e.g. McLinden et al., 2000; McCormack et al., 2006).

For all this, we think it is appropriate to refer to the BMS ozone model as a new model.

**Comment #2. Since one of the main points of the study is to demonstrate the improvement of the new ozone tracer compared to the default approach, more details are needed about the exact differences between the BMS and CD approaches. Therefore, while it may be true that some of the details that are wanting can be found in Monge-Sanz (2011) I feel that a subset can be repeated here. The reason for this is that the authors tend to attribute most of the improved ozone performance to the incorporation of implicit heterogenous chemical effects but there may be other factors at play as well. In particular, what is the vertical resolution of the 2D photochemical model in the default ozone implementation? The coefficients for the BMS scheme are derived from the box model version of a 3D CTM. They are calculated at 24 levels (and then presumably interpolated somehow to the native vertical grid of the IFS forecasts, although this detail is not provided). Are the coefficients from the default ozone linearization also available at these levels? If the treatment of vertical gradients is still more simple in that model it is not obvious to me that the BMS and CD main differences reside in the treatment of heterogeneous chemistry. Please, I strongly suggest the authors add more details to Section 2.1 that address resolution (and forcing set) differences between the box models separately used to produces the coefficients for these**

**different ozone tracers. Moreover, speculate on how these aspects of the ozone tracer linearization may be contributing to the differences seen (in the conclusions).**

We respond to this comment in two parts, related to each of the aspects the Reviewer mentions (resolution and forcings).

Regarding vertical resolution, the CD scheme original number of levels is actually larger (60 levels) than that of the original BMS number of levels (24 levels). Both schemes are then interpolated to the same grid for each experiment in the ECMWF model, using the same interpolation method for both. If the original number of levels were playing a more important role than the different representation of the heterogeneous chemistry, we should see deterioration when using the BMS model option instead of the observed improvement. This, and the fact that the largest differences between schemes are found in high latitudes, followed by those in mid latitudes, and only small differences over low latitudes, supports that the different treatment of heterogeneous chemistry is the most significant factor in the performance of the scheme.

We now add information on the differences in original resolution of the coefficients in both schemes into Section 2.1.

**#2(ctd): Another point that needs clarification in Section 2.1 is the meteorological fields that are used to initially derive the coefficients in both the BMS and default ozone linearization. What is the limitation of using coefficients based on box model runs using input meteorology for one representative year? If the results are based on coefficients on a year with mild temperatures one might expect that this would restricts the ability of the scheme to reproduce realistic ozone loss during colder winters and vice versa. Is this an issue for both ozone tracers?**

The box model used to derive the coefficients for the BMS scheme in this study was initialised from a 3D SLIMCAT full-chemistry reference run forced by ERA-40 meteorological fields, corresponding to year 2000 (more details about the use of one particular year in later paragraph here below). While the 2D model used to derive the CD coefficients was run from output from the climate model ARPEGE averaged for the period 1990-2000 (Cariolle and Teyssèdre, 2007).

In the case of the BMS scheme, the use of meteorology from one particular year was thoroughly addressed by Monge-Sanz et al. (2011). In their study, they compared two different versions of the parameterisation, one was based on a representative year with cold winter conditions (year 2000), and another version based on a year with mild winter conditions (year 2004). Their results demonstrated that when the coefficients are derived from years with more extreme conditions (cold winters), the scheme is able to realistically capture ozone distributions for both cold and mild conditions. Results illustrating this were shown in Figures 10, 11 and 12 in Monge-Sanz et al. (2011).

In the case of the CD scheme, the coefficients are derived from the mean values from a climate model run averaged over a 20-year period. This averaging process already unrealistically smooths away extreme ozone events, and the adhoc heterogeneous term in the CD scheme, depending only on a temperature threshold for a geographically restricted region, is not realistic enough to adapt to different meteorological conditions.

We have added several sentences to clarify these points in the manuscript (end of Section 2.1).

**Comment #3. Major Comment: One issue that is not discussed is the increased computational expense of the BMS scheme, which is introduced by needing to run the full 3-D CTM (and feed into the box model of the CTM to produce the coefficients). While this may be only a "one time" step it nonetheless is more expensive than simply running a 2-D box model using, for example, observed ozone climatologies that already exist. Indeed, the authors even suggest that it may be needed to constantly run the full CTM in order to provide a new "carefully chosen climatology term…from an updated run of the parent full-chemistry CTM." While that certainly will be good (to the extent that the CTM itself does a good job at reproducing observed ozone) it is nonetheless not a small computational ask. I suggest that the authors discuss this trade-off between performance and computational cost in the conclusions.**

We have not discussed the computational costs of the schemes because it is not a factor that affects the runs we perform and evaluate in this study. Both the BMS and the CD schemes have been derived off-line and for multiple applications.

Although a 3-D CTM is more computationally expensive than a 2-D CTM, the 3D CTM simulations we use are offline reference runs and serve multiple purposes, i.e. they exist independently of whether they are used to derive an ozone parameterisation or not. The cost of the box model runs required to obtain the coefficients were reasonably low (i.e. could run on a local cluster). And also please take into account that the forcing files used by the 2D photochemical model came from a chemistry-climate model, which also increases the computational cost of that scheme (although, also here the cost of the CCM simulations did not affect the NWP runs).

Since neither the box model nor the CTM are coupled to the NWP model we use, the computational cost of the offline simulations is not affecting the NWP model performance. The only implied cost in the NWP model is that of the one tracer needed to simulate ozone, both in the experiments using the CD and the BMS ozone schemes; reading-in one fewer coefficient in the case of the BMS scheme. Thus, the computational cost in the NWP model is almost identical in both cases, slightly lower cost in the BMS case.

To provide an improved climatology term, it is not necessary to run the CTM constantly, but only when new developments implemented within the CTM have an impact on the O3 climatology term. In our case we know that the tropical bias has been solved in recent versions of the CTM, and an updated version of the BMS scheme can be derived in which the tropical bias exhibited in the version used in our study will not be present. This is what we already discussed in Section 5.2 (original lines 475-480). To make it clearer, we now add a reference in Section 3.2.1 (the one quoted in this Reviewer's comment) pointing to the updated climatology discussion in Section 5.2.

**Minor Comments:**

**Line 29: A semi-colon (not comma) is needed after "exothermic reactions".**

This sentence has been edited to improve readability.

**Line 99: A "the" is needed in "can simulate time".**

Done, thanks.

**Line 103: More description here is needed regarding the "tendencies derived from the fullchemistry CTM runs". My specific concern regards ambiguities about the forcings and boundary conditions used for the CTM runs from which these full tendencies are derived. In particular, over what time period are these CTM runs performed? More generally, what are the boundary conditions and compositional forcings used in the CTM simulations and how consistent are these with those used in the IFS forecast ensembles?**

To derive the tendencies, the box model used to derive the coefficients for the BMS scheme was initialised from a 3D SLIMCAT full-chemistry reference run forced by ERA-40 meteorological fields, corresponding to year 2000 (more details about this in our response to a previous comment above).

The BMS model scheme, and the CD scheme as well, are not exclusively designed for the IFS model. In particular for the BMS scheme, we emphasize its portability to any global model that needs to simulate realistic stratospheric ozone without the computational costs and technical requirements of full-chemistry. To enable such portability, the simplified model needs to be adaptable, robust and consistent in itself.

The way the coefficients are computed, using sensitivity runs of the full-chemistry box model, ensures that such tendencies depend on temperature differences and not on absolute temperature values, which provides robustness to the method, i.e. depend more on the full-chemistry reactions and rates included in the box model than on the forcing fields. Thus, the tendencies coefficients can cause no inconsistency. The argument for the climatology terms is slightly different, in particular the temperature climatology term can differ from the temperature field in the ECMWF runs. This is one of the reasons why considering the use of climatology terms based on observations cannot be ruled out for future implementations, but for the sake of robustness we decided to use both schemes as they are originally derived. Also please note that the climatology terms in the ozone scheme are not passed to the ECMWF model, but used only as reference to compute ozone tendencies.

For the ozone field in the ECMWF runs, we ensured that ozone values used in the initialisation of the ECMWF runs did not cause any issue by allowing a spin-up period at the start of the IFS model experiments.

Full details about the CTM runs, forcings and periods considered are fully described in Monge-Sanz et al. (2011), and we have now also added more details to this part of the text (last paragraph in Sect.2.1) to clarify all points related to this comment, and also in response to one related previous comments above.

**Line 122: The "to" after "sunlight" should be removed.**

OK, done.

**Line 133: I'm not sure I agree with this statement beginning with "In addition, the coefficients…". While it may be true that the coefficients are derived from a 3D model, what is used in the BMS scheme are the zonally averaged fields. mPlease reconcile this discrepancy.**

We think there is no such discrepancy. The zonal average of our 3D coefficients may smooth longitudianal variability, but the coefficients from the 2D model simply ignore any longitudinal variability. In this respect the level of realism achieved by a 3D model, even when zonally averaged, is superior to that of a 2D model. In addition, the BMS scheme always provides the possibility to use the full set of 3D coefficients, when computational costs in the global model allowed.

**Line 141: I am not sure I agree with the sentence "The IFS configuration in each pair … differs only in the scheme used to model stratospheric ozone." Is this really true? Later on in the manuscript, you write that the old scheme does not include radiative feedbacks from ozone so isn't this a key difference with the new BMS scheme? I understand that you ran experiments with the BMS scheme in which feedbacks are/are not incorporated but please indicate here that this is another key difference between the schemes.**

In this sentence we meant that we only change the representation of ozone in the ECMWF experiments. There is no other change in the model configuration, i.e. all other physics, parameterisations, numerics, resolution etc is the same.

To be precise the ozone schemes we use are only two: the CD and the BMS schemes. But we agree with what you say, as it is also true that we use the BMS as prognostic ozone interactive with radiation in several runs.

We have edited the sentence to avoid misunderstanding, it now reads: "*…differs only in the scheme used to model stratospheric ozone, and whether prognostic ozone is used or not by the radiation scheme. All other aspects of the model configuration remain identical.*"

**Line 145: Given that the box model coefficients are inferred from a much coarser CTM output (7.5 horizontal resolution and 24 levels, according to Monge-Sanz et al. (2011)) how do you account for the increased resolution of the native model grid? Do you simply interpolate? Wouldn't it be worth running the CTM at a higher resolution (albeit with the additional expense) to get more realistic coefficients?**

Ideally, yes, but in practice NWP resolutions are very demanding for the full-chemistry CTM runs required. And producing CTM runs to compute coefficients at all the possible resolutions IFS can be run would be impractical.

That is why we obtained the coefficients at the resolution that was standard for the off-line CTM multiannual full-chemistry runs at the time. This also allowed us to fairly compare within the CTM the performance of the parameterisation and the ozone field from full-chemistry.

Work is ongoing to produce an updated version of the BMS scheme which, among other developments, will be computed at higher resolution. In any case, the CTM cannot reach the operational resolution of IFS for the length of the full-chemistry runs required, and some spatial interpolation will still be required.

The same interpolation method is used for both ozone schemes to ensure numerical consistency. First, linear interpolation in pressure is performed followed by linear interpolation in latitude. We applied this for the different vertical level configurations in IFS; this is the usual practice at ECMWF for all parameterisations that are sensitive to the vertical discretization.

We have added information about this to the main text in Section 2.1 when describing the original resolution of the ozone schemes, and in Section 2.2 when describing the experimental setup resolutions in IFS.

**Line 164: Remove "an" before "horizontal resolution".**

Done, thanks.

**Line 166: Are these ensembles also launched in May and November of each year over the period 1981-2010? It is not clear when these are initialized.**

Yes, these runs have the same start dates and integration period as the other ensembles. We have added this information into the text now.

**Line 171: An "and" is needed before "the National Oceanic …".**

Thanks, this has been added.

**Line 185: Experiment names (from Table 1) need to be indicated as it is not clear if the only difference here is the linearization scheme or the interactions with the radiation. Simply providing the experiment names will help clarify this. In particular, are you comparing exp001bms and exp001cd or exp002bms and exp001cd?**

When comparing the performance of the two ozone schemes in terms of the ozone distribution they provide, we compare them in the fairest possible way: both of them non-interactive with radiation. This Figure shows exp001bms and exp001cd, we have now added this information into the text to make it clearer.

**Line 200: Typo in "characterstics".**

Thanks, this is now corrected.

**Line 209: What do you mean exactly by "the more realistic link with temperature in the new scheme"? Is this because of the implicit incorporation of heterogenous chemistry in the coefficients or are you comparing the schemes with and without feedbacks to radiation? Again, clarification about which exact experiments are being analyzed is needed. See my previous comment.**

Yes, it is because the new scheme, by incorporating both gas-phase and heterogenous chemistry in a consistent way, simulates more realistic links with temperature than a scheme that artificially separates heterogeneous chemistry using a fixed adhoc T-threshold term. In reality the temperature threshold for the formation of polar stratospheric clouds (PSCs) depends on altitude and trace gas concentrations.

Here we are analysing experiments in which the schemes are not linked to radiation (exp2002bms and exp2002cd in Table 1). We now add this information into the text.

**Line 216: Is there a reason why these experiments, which are described earlier, are being described again? Why the need to repeat these experiment details? Or are these different runs?**

You are right, these are the experiments described earlier. It was text remaining from previous manuscript versions. We have now edited this part of the text to avoid confusion. Thanks.

**Line 229: I'm not sure "manifests" is the right word here.**

Now changed to "indicates".

**Line 261: I'm not convinced that the use of the "a recent observation-based climatology" would necessarily be ideal. Isn't one of the main improvements afforded from the BMS scheme the fact that it better ensures internal consistency between the gas phase and heterogenous chemistry? To this end, therefore, wouldn't replacing the climatology term with one derived from observations break consistency with the other terms (which are derived from the CTM)? If consistency is a desirable feature, I would think that using a combination of model and observationally derived coefficients would be problematic.**

We agree that this is not the ideal option and that the gains would clearly need to overweight the relative inconsistencies it may introduce. But for NWP models it is an option that can be considered to improve known biases in some regions or in some model versions.

The tendency coefficients in the ozone model would remain realistic and consistent among them, and would also continue to include both heterogeneous and gas-phase chemistry effects. For the reference/climatology terms, if they came from a climatology based on observations, they would also indeed incorporate both heterogeneous and gas-phase chemistry effects. Of course, all climatology terms would need to be as consistent as possible among them, i.e. climatology for the ozone term and for the partial column term would need to come from the same observations.

We have added some text after line 261 (original numbering) to clarify that we do not consider this option as ideal, but as a reasonable option to enhance the performance of the ozone model if needed when known biases are present in the scheme's climatology terms. Text added: "*Although the use of an observation-based reference term is a reasonable option to improve known biases in some regions, one needs to be careful to avoid degrading the scheme's performance in other regions and to keep internal consistency of the ozone scheme as much as possible (see additional discussion on this point in Section 5.2).*"

**Line 285: Again, why can't this comparison be cleaner? You should do either BMS w/wout radiative feedback (or using CD). What does that show? I suggest the authors replace this text here with a discussion of those results since this is not a proper apples-to-apples comparison.**

The first part of the paper demonstrates that the BMS scheme provides more realistic ozone distributions compared to observations than the CD scheme, and we have done this using, as you suggest, clean pairs of experiments. The rest of the paper focuses on case-studies and comparison of the best performing ozone model from the initial comparison (the BMS scheme) against the default

option used in the operational model (climatology passed to the radiation scheme). Please note that assessing the new model option for operational purposes is a major aim of this paper, which is why once we identified the improved model, then the focus is on comparing its suitability against the default operational option. The CD scheme had been implemented in the IFS a long time ago but because of intrinsic deficiencies mainly related to its treatment of heterogeneous chemistry, its use in operations had not been accomplished. That is why our main purpose, once we have verified the improvements achieved by the new option (BMS model), is to compare the new BMS prognostic ozone with the default configuration of this IFS model version (which uses the ozone climatology instead of a prognostic ozone model).

**Line 480: This is a more of comment that could be addressed in the conclusions but have you considered modifying BMS coefficient calculation to account for zonal variations in ozone? If this is what is already meant by "the BMS scheme coefficients could be provided in a 3D grid" then please clarify that, indeed, this is the case.**

Yes, this is what we meant by "provided in a 3D grid". The scheme already accounts for latitudinal variations in ozone chemistry, as coefficients are provided as a function of latitude. But to make full use of the advantages of the 3D full-chemistry runs we used, we plan to test a version that also includes longitudinal variations in the coefficients. This is indeed part of the ongoing work we are doing for the updated version of the BMS scheme. We have edited that sentence in the main text to make it clearer. Thanks.